# GNSS-RO Deep Refraction Signals from Moist Marine Atmospheric Boundary Layer (MABL)

**Dong L. Wu** [1,*], **Jie Gong** [2] and **Manisha Ganeshan** [3]

1   NASA Goddard Space Flight Center, Greenbelt, MD 20771, USA
2   GESTAR-II, University of Maryland, Baltimore County, Baltimore, MD 21228, USA; jie.gong@nasa.gov
3   GESTAR-II, Morgan State University, Baltimore, MD 21251, USA; manisha.ganeshan@nasa.gov
*   Correspondence: dong.l.wu@nasa.gov

**Abstract:** The marine atmospheric boundary layer (MABL) has a profound impact on sensible heat and moisture exchanges between the surface and the free troposphere. The goal of this study is to develop an alternative technique for retrieving MABL-specific humidity ($q$) using GNSS-RO data in deep-refracted signals. The GNSS-RO signal amplitude (i.e., signal-to-noise ratio or SNR) at the deep straight-line height ($H_{SL}$) was been found to be strongly impacted by water vapor within the MABL. This study presents a statistical analysis to empirically relate the normalized SNR ($S_{RO}$) at deep $H_{SL}$ to the MABL $q$ at 950 hPa (~400 m). When compared to the ERA5 reanalysis data, a good linear $q$–$S_{RO}$ relationship is found with the deep $H_{SL}$ $S_{RO}$ data, but careful treatments of receiver noise, SNR normalization, and receiver orbital altitude are required. We attribute the good $q$–$S_{RO}$ correlation to the strong refraction from a uniform, horizontally stratiform and dynamically quiet MABL water vapor layer. Ducting and diffraction/interference by this layer help to enhance the $S_{RO}$ amplitude at deep $H_{SL}$. Potential MABL water vapor retrieval can be further developed to take advantage of a higher number of $S_{RO}$ measurements in the MABL compared to the Level-2 products. A better sampled diurnal variation of the MABL $q$ is demonstrated with the $S_{RO}$ data over the Southeast Pacific (SEP) and the Northeast Pacific (NEP) regions, which appear to be consistent with the low cloud amount variations reported in previous studies.

**Keywords:** atmospheric boundary layer; specific humidity; diurnal variation; GNSS-RO; signal-to-noise ratio; deep refraction; grazing reflection

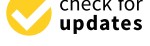



## 1. Introduction

The atmospheric boundary layer (ABL), an interface between the free troposphere and the surface, plays a key role in regulating exchanges of sensible heat and moisture fluxes through turbulent processes, as well as in the transport of aerosols, pollutants, and greenhouse gases [1]. In the case of marine ABL (MABL), cloud formation and atmosphere–ocean interactions are determined by key variables such as ABL height (ABLH) and water vapor mixing ratio ($q$) abundance. Because of large ABL variability and complex processes in this shallow layer, lack of all-sky $q$ and ABLH measurements on a global basis means poor constraints on model parameterization schemes and hinders the understanding and representing of these processes in global climate models (GCMs) [2–4]. As a result, large uncertainties remain in current GCMs and their low-cloud feedback processes [5].

It has been a great challenge to remotely sense MABL $q$ from space, not only because MABL is a shallow layer but also because clouds often prevent sensors from penetrating into this layer. While the total column of water vapor (CWV) can be measured well with satellite microwave (MW) [6] and shortwave-IR (SWIR) [7] sensors, the $q$ observations within the MABL are still limited. Millan et al. [8] attempted to use the differencing between MW-CWV and SWIR-CWV above cloud to infer MABL $q$, but the technique requires accurate knowledge of MABL cloud top and temperatures. Thus, the inferred MABL $q$ with this differential technique is prone to SWIR-CWV errors above clouds.

Global Navigation Satellite System Radio Occultation (GNSS-RO) has emerged as a promising new technique to remotely sense MABL on a global all-weather basis. Because GNSS-RO can penetrate clouds from a rapidly growing number of SmallSat/CubeSat constellations, the technique has the advantage of providing the needed global spatiotemporal sampling to better characterize the MABL $q$ and its variability. The GNSS-RO technique offers a good (~200 m) vertical resolution and sensitivity to the ABL top associated with a sharp vertical gradient in moisture/temperature profiles, which produce a measurable signal in the bending angle ($a$) or refractivity ($N$) profile. The gradient method for the $a$ or $N$ profile has been successfully applied to GNSS-RO data to measure ABLH [9–12].

However, MABL $q$ measurements are still lacking, because only a small fraction of the $a$ or $N$ profiles in GNSS-RO operational Level-2 products can reach below ABLH to provide the MABL $q$ measurement. Approximately 50% of the RO profiles from COSMIC-1 (Constellation Observing System for Meteorology, Ionosphere, and Climate-1) can reach to 1 km above the surface in the tropics, and less than 10% down to 0.5 km. It requires a good signal-to-noise ratio (SNR) from GNSS-RO to produce the useful excess phase measurements, of which the latter is essential to retrieve the $a$ or $N$ profile. Because the lower tropospheric RO SNR decreases with height due to atmospheric defocusing effects, the number of useful excess phase measurements for the $a$ or $N$ retrievals drops substantially as the signal becomes noisy. As a result, most of the GNSS-RO water vapor measurements and intercomparison studies have been limited to the regions above the ABL at pressures <900 hPa or over landmasses where radiosonde observations are available from the routine operation [13–15]. In addition, the RO $N$ retrieval from the Abel inversion is known to have a low bias in the lower troposphere, which leads to a dry bias in the derived MABL $q$ [16,17]. It remains as an active research topic to understand its root cause and develop correction methods. Because of these limitations, the GNSS-RO $a$ or $N$ data in the lower troposphere (<8 km) have not been used in most of the global forecast and data reanalysis systems [18].

Motivated by the need for more accurate MABL $q$ observations, the goal of this study is to develop an alternative technique for MABL $q$ using the GNSS-RO measurements from deep-refracted signals. Here we focus on the relationship between the MABL $q$ abundance and the SNR at straight-line heights ($H_{SL}$) below $-80$ km where the RO receiver is far into the occulted situation. We found that the SNR from deep $H_{SL}$ is correlated strongly with the MABL $q$ abundance at 950 hPa (or 400 m) above the sea level ($Z_s$). To better understand the causes and limitations of deep $H_{SL}$ SNR signals, we investigated the signals from different $H_{SL}$ levels as well as the potential applications of these deep $H_{SL}$ SNRs in MABL $q$ data assimilation.

## 2. Method and Data Analysis

Systematic studies of GNSS-RO SNR from deep $H_{SL}$ are enabled by the now-widely used open-loop (OL) operation since successful experiments on planetary [19] and terrestrial atmospheres [20]. The OL operation allows the RO signal to be tracked continuously at deep $H_{SL}$ even where SNR becomes noisy or experiences a temporary loss. Unlike a closed-loop (CL) operation, in which a phase-locked loop (PLL) circuit is used to track the signal frequency based on previous measurements, in an OL operation, the receiver relies on a modeled reference signal frequency from orbit dynamics, receiver clock drift, and estimated atmospheric bending effects to track the anticipated RO signal regardless of SNR fluctuations. The CL operation works well in cases where SNR is strong, but a long period of low SNRs would prevent the PLL from updating the reference signal and result in a loss of tracking the GNSS signal. The introduction of the OL operation is meant to reduce the number of loss-of-tracking incidences so as to improve the sampling at low $H_{SL}$. In most current GNSS-RO operations, the transition from CL and OL occurs approximately at $H_{SL} = -20$ km, and the hybrid CL-to-OL operation is implemented for both rising and setting RO soundings.

*2.1. GNSS-RO Bending in a Moist Atmosphere*

Although both RO SNR and phase measurements contain information about MABL properties, the SNR-based techniques are least explored in GNSS-RO research. Because GNSS-RO SNR is a radiometric signal, a proper calibration scheme, in either the absolute or relative sense, must be established before RO radiometry can be applied for physical variable retrievals. It remains as an active research area to fully understand how a thin atmospheric layer refracts the transmitted radio signal down to these deep $H_{SL}$ in the RO sounding. In a research monograph, Melbourne [21] provided a comprehensive review on spaceborne GNNS-RO fundamentals, system error sources, and diffraction effects associated with the RO signal. Like Fresnel diffraction from an obstacle's edge, the Earth's limb and a sharp MABL can produce a diffractive effect on the RO signal. This diffractive effect can extend the signal below the sharp edge of the obstacle but with a limited depth. It requires both refractive and diffractive processes in the radio wave propagation to produce the RO signal at deep $H_{SL}$. In a simulation study using the multiple phase screen (MPS) model [21], sensitivity analysis confirmed that the diffraction and refraction from a thin moist layer can strongly impact characteristics of RO signals at deep $H_{SL}$, as predicted from the analytical solution [21]. Sokolovskiy et al. [22] also showed that the deep $H_{SL}$ signal depends not only on the vertical gradient of MABL but also on the horizontal extend of the stratiform layer.

The GNSS-RO phase-based technique, which is employed in the operational Level-2 data processing, determines the atmospheric refractivity profile from bending-induced excess phase measurements. The excess phase is the difference between straight-line distance and bended/delayed wave propagation length. The refractivity $N$ of radio wave propagation can be modulated by atmospheric pressure ($P$), temperature ($T$), moisture pressure ($P_w$), and ionospheric electron density ($n_e$), as in Equation (1)

$$N = (n-1) \times 10^6 = 76.6 \frac{P}{T} + 3.73 \times 10^5 \frac{P_w}{T^2} - 4.03 \times 10^7 \frac{n_e}{f^2} \tag{1}$$

where $n$ is the atmospheric/ionospheric refractive index, $P$ and $P_w$ are in hPa, $T$ is in K, $n_e$ is in m$^{-3}$, and $f$ is the RO frequency in Hz. Water vapor-specific humidity $q$ is related to $P$ and $P_w$ in $q \approx 0.622 \cdot P_w / (P - 0.378 \cdot P_w)$. As shown in Figure 1, the vertical gradient of the atmospheric $N$ profile plays a critical role in the bending effect on the GNSS-RO signals. Any sharp $N$ gradient can induce strong scintillation, ducting, and diffraction effects, causing a temporary loss of RO signals. As summarized below, these effects are common in the GNSS-RO observations in the lower atmosphere:

- *Normal bending*: As radio waves pass through the atmosphere, their paths are bended by the atmospheric refractivity $N$ profile with a vertical gradient. This vertical $N$ gradient acts as an optical prism to refract and diverge the RO beam, causing a weaker power per receiving area compared to the free-space case. This is known as the *defocusing effect* [17]. Thus, the RO signal amplitude, or SNR in voltage (V/V), decreases gradually at lower $H_{SL}$ due to the *defocusing effect*. In the presence of a sharp $N$ vertical gradient, such as the ABL top, a threshold may be used to detect ABLH [9,23].

- *Grazing reflection*: Unlike GNSS-R, which acquires GNSS signals at a larger reflection angle, grazing reflection takes place at a bending angle similar to the RO sounding. The reflected signal at a grazing angle does not necessarily reverse polarization, which allows the direct and reflected signals to interfere with each other at the RO receiver. Thus, grazing reflection is a multi-path problem in GNSS-RO observations. The interference generates coherent or semi-coherent signals that manifest themselves in a radiohologram of the RO sounding. The reflective surface can be either a smooth ground surface or an elevated atmospheric layer (e.g., water vapor layer). Both RO grazing reflection and GNSS-R signals can be used to detect smooth, flat surfaces [24,25].

- *Super-refraction (SR) effect*: This occurs where atmospheric bending exceeds the Earth's curvature ($dN/dz < -157$ N-unit/km). In such cases, the RO path is bended down more than the Earth's curvature such that the signal is unable to reach the RO receiver [26]. However, the SR condition is sensitive to the angle of incidence with respect to the refractive surface in such an impact. As the GNSS-RO sounding progresses, especially during the open-loop operation, the incident angle would vary with $H_{SL}$ and allow the RO signal to re-emerge from a temporary loss. The re-appearing signals are still partially affected by the SR interface and can cause a systematic negative bias in the $N$ retrieval (i.e., negative $N$-bias) [17].

- *Ducting effect*: The RO propagation can be trapped in a thin atmospheric layer or between the surface and a reflective atmospheric layer (e.g., ABL top). Under a proper refraction–reflection condition, the layer can act as a duct to channel the RO propagation to reach a farther distance. Sokolovskiy et al. [22] studied the RO ducting effect with numerical model simulations and found that the length of ducting could play a key role in the deep $H_{SL}$ RO signals. The longer the duct length, the deeper the RO signal can reach. The simulations also showed that ducting can induce a temporary loss of RO signals or a U-shape in the SNR profile due to multi-path interference. The U-shape essentially bifurcates the RO signal, as illustrated by two blue lines Figure 1, and induces a minimal SNR before the signal re-emerges at a lower $H_{SL}$.

- *Diffraction effect*: Like the light passing through a sharp edge object, the RO signal also experiences diffraction effects as it passes through a sharp layer such as ABL. The diffraction fringes are present in the RO signal amplitudes at deep $H_{SL}$, as simulated from a sharp ABL [17–21,27]. Consequently, modeling with wave optics must be employed to comprehend the SNR signals in these situations. The RO diffraction effect is more pronounced in the object without an atmosphere, such as the lunar occultation [28], but it is often overwhelmed by the other aforementioned effects when an atmosphere is present.

In reality, complex atmospheric moisture layers can produce a mixed effect in the RO sounding from all the above. It is often difficult to distinguish one from the other in an observed RO profile. As a result, sophisticated tools such as radiohologram and canonical transform are employed to diagnose the GNSS-RO measurements and improve the RO retrievals. Radio holography is useful to identify coherent oscillations within an RO sounding, such as the reflection-induced interference features. The canonical transform method allows a reconstruction of multi-path propagation from the full-spectrum inversion that makes use of both the RO phase and amplitude profiles. Nevertheless, a number of these tools involve tedious, time-consuming calculations and are impractical as an operational algorithm for processing a large-volume of data.

*2.2. GNSS-RO Radiometry*

In this study we analyzed multi-years of the Level-1B GNSS-RO data (e.g., atmPhs and conPhs) published at CDAAC (COSMIC Data Analysis and Archive Center) and focused on the signal amplitude (i.e., SNR) at $H_{SL} < -80$ km. Instead of using the standard Level-2 products derived from the excess phase ($f_{ex}$), we sought an alternative approach for the MABL $q$ that could overcome the sampling and bias problems associated with the $f_{ex}$ measurement. One of the advantages of using the SNR measurement is its availability in every RO sounding at very low $H_{SL}$, as long as MABL $q$ can produce a detectable signal from refraction. In the cases where the SNR is weak and the $f_{ex}$ measurements are too noisy to produce the useful bending angle retrieval, the SNR data are still available and can be used to infer the MABL $q$. Thus, this study was motivated to establish a robust relationship between deep $H_{SL}$ SNR and MABL $q$ so that a retrieval algorithm could be developed.

To properly analyze the deep $H_{SL}$ RO signal amplitude for radiometry, we first normalized the L1 ($f = 1.57542$ GHz) SNR profile by its value in the free atmosphere (SNR$_0$) to account for large profile-to-profile SNR variations in RO observations (Appendix A). The self-sufficient normalization divides each SNR profile by its SNR0 derived from

$H_{SL}$ = 40–60 km, where the atmospheric bending is negligible. This normalization provides a relative radiometric calibration with respect to its power in the free atmosphere. Because the RO SNR can vary with the transmitter power and receiver antenna pattern (Appendix A), the normalized SNR profile greatly reduces these effects, so that it can be used to infer atmospheric properties. In addition to $SNR_0$ variations, the RO SNR measurement noise ($\sigma$) has a secondary effect on radiometry and must be taken into account in the normalization. As an RO receiver-dependent variable, it is subtracted from the SNR profile based on the RO receiver–transmitter pair (Table A1 in Appendix A). The receiver noise effect becomes noticeable in analyzing the weak signal at deep $H_{SL}$. The final normalized SNR ($S_{RO}$) is given by,

$$S_{RO} = (SNR - \sigma)/(SNR_0 - \sigma) \tag{2}$$

The SNR noise ($\sigma$) is determined individually for each group of GNSS-RO measurements from very deep $H_{SL}$. More detailed discussions can be found in Appendix A, and the measurement noise from various GNSS-RO pairs are listed in Table A1.

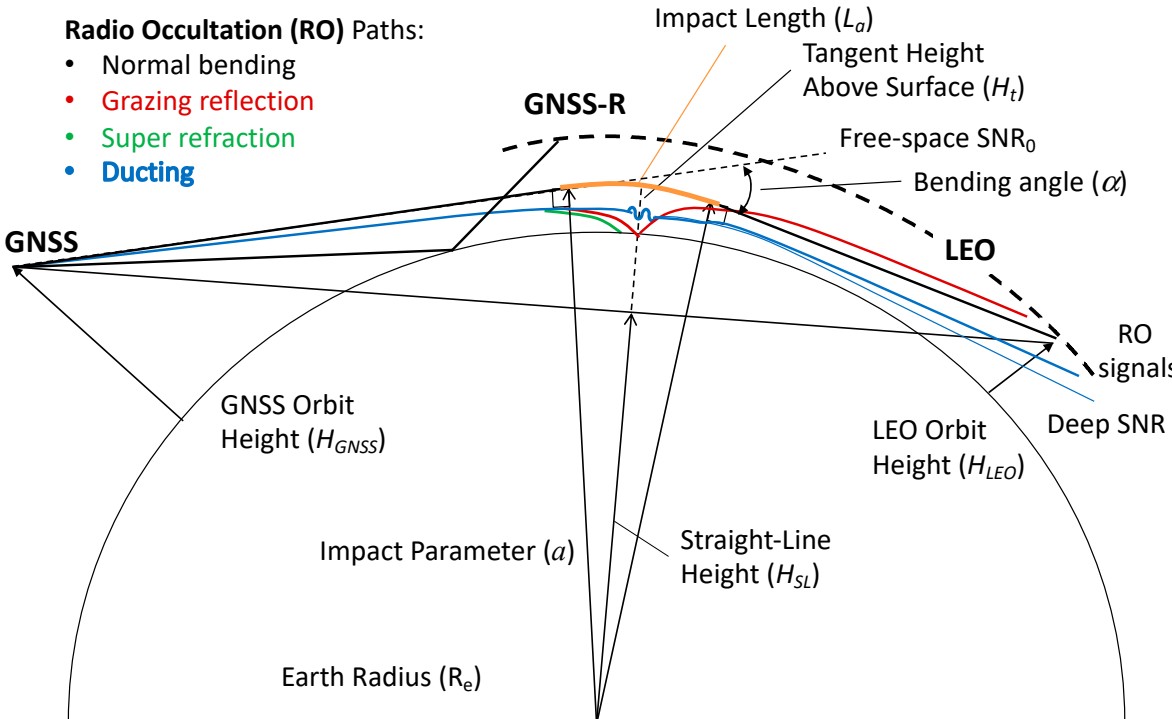

**Figure 1.** Schematic diagram to illustrate GNSS-RO observation geometry with several key bending scenarios in color for normal bending, grazing reflection, super refraction, and ducting conditions. The orange line in the middle of the ray path denotes the section where radio wave propagation is significantly impacted by the atmosphere. The horizontal extent of this section is impact length ($L_a$). The refraction and diffraction from the water vapor layer ducting are most likely to produce an RO signal at a deep straight-line height ($H_{SL}$) or large bending angle ($a$). The RO receiver signal amplitude (signal-to-noise ratio or SNR), impact parameter ($a$), LEO satellite height ($H_{LEO}$), GNSS satellite height ($H_{GNSS}$), and tangent height ($h_t$) are also defined in the diagram.

In addition to the mean $S_{RO}$ value, $S_{RO}$ scintillations in each profile contain valuable information on layered structures in the atmosphere. As discussed above, a sharp vertical gradient in $N$ can induce a strong response in refraction/defocusing/interference effects associated with large scintillations near the height where the gradient occurs. To characterize the scintillation intensity, we computed a profile of the 1 s $S_{RO}$ variance from the difference between $S_{RO}$ and its 1 s running mean ($\overline{S_{RO}}$), as defined below,

$$\sigma_S^2 = var\left(S_{RO} - \overline{S_{RO}}\right) \tag{3}$$

Like $S_{RO}$, $\sigma_S^2$ is a function of $H_{SL}$, but the vertical resolution of the 1 s average may differ from occultation to occultation, depending on relative motions between GNSS and LEO satellites. Typically, it varies between 1 and 2 km. Thus, as the RO passes through horizontally stratified atmospheric layers, strong scintillations may occur to enhance $\sigma_S^2$.

The zonal mean climatology of $S_{RO}$ and $\sigma_S^2$ from January 2008 reveals several interesting features induced by atmospheric layered structures (Figure 2) The lower atmospheric $q$ (primarily from the ABL) is responsible for the enhanced $S_{RO}$ and $\sigma_S^2$ seen at $H_{SL} < -80$ km. The RO signals extended below $H_{SL} = -100$ km, mostly from the moist tropical atmosphere, are far below the cutoff height in the Level-2 bending angle retrievals. The gradual decreasing $S_{RO}$ with $H_{SL}$ is a manifestation of the defocusing effect from the atmosphere. The bulk of the $S_{RO}$ enhancement is biased slightly towards the summer hemisphere where there is more solar isolation and water vapor abundance in the subtropics. A similar $S_{RO}$ enhancement is evident at $H_{SL} = -10$ km for mid-to-low latitudes, with a sharp transition at ~55° S and ~45° N. This height ($H_{SL} = -10$ km) is above the CL-to-OP transition, which typically occurs at $H_{SL} = -20$ km. Near the tropopause, the $S_{RO}$ exhibits a low value, likely due to the defocus effect caused by the sharp vertical temperature gradient.

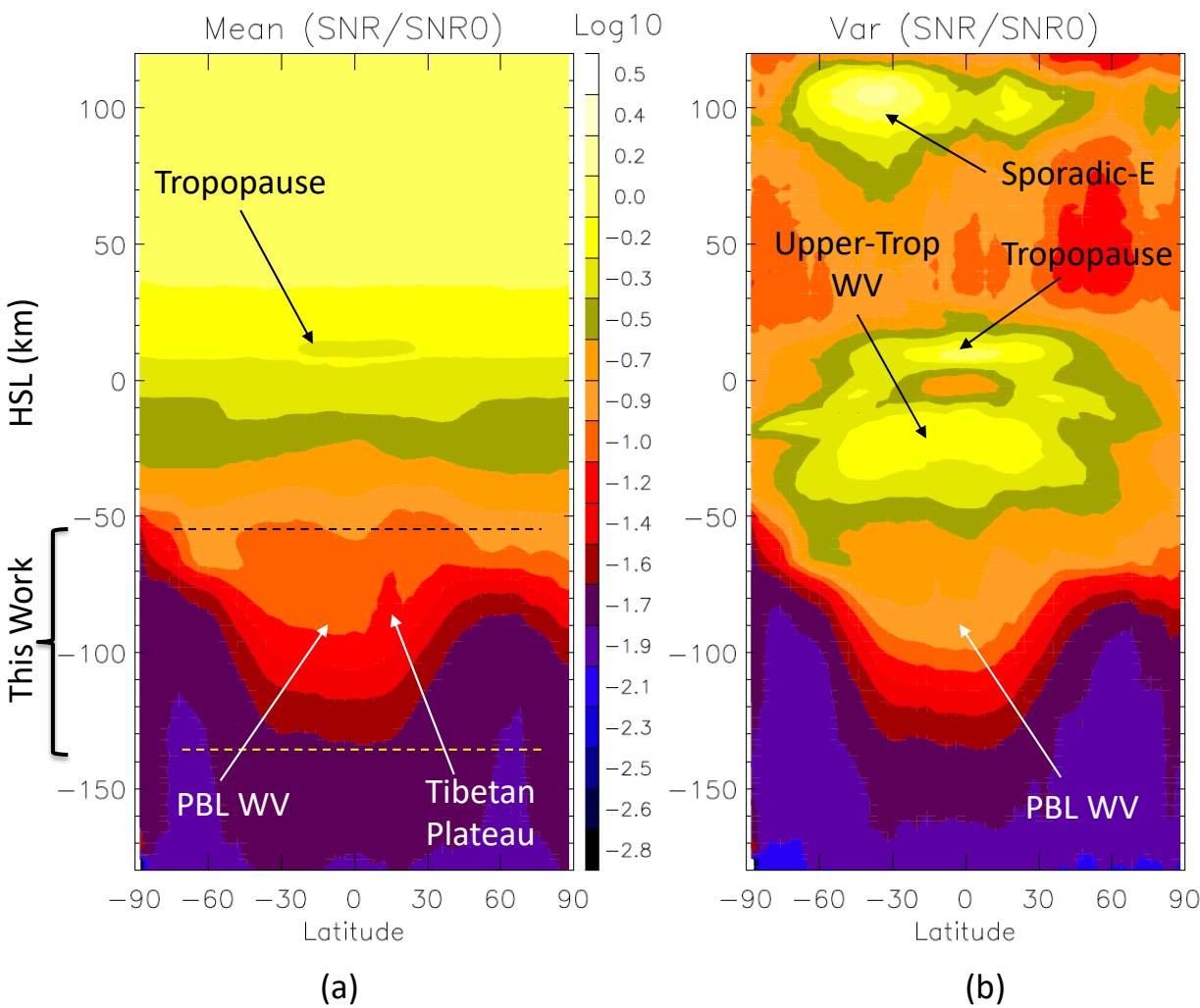

**Figure 2.** (**a**) Zonal mean $S_{RO}$ and (**b**) $\sigma_S^2$ from COSMIC-1 for January 2008 as a function of latitude and $H_{SL}$, showing enhanced SNR at deep $H_{SL}$ from ABL $q$ and increased variances from layered regions. A significant reduction from the Tibetan Plateau is reflected in zonal mean $S_{RO}$.

The GNSS-RO vertical resolution, defined by the first-Fresnel-zone dimension, varies from ~1.5 km in the stratosphere to 0.1–0.5 km in the lower troposphere [29,30]. Such a

high-resolution sounding makes the technique very sensitive to atmospheric thin layers with a sharp vertical refractivity gradient. The regions with a sharp vertical gradient can be readily seen in Figure 2b from the variance $\sigma_S^2$. The most prominent regions with thin layers include the tropopause, upper-tropospheric water vapor, and sporadic-$E$ ($E_s$) in the ionosphere.

### 2.3. Geometric and Wave Optics

To fully understand the deep $H_{SL}$ GNSS-RO signals from MABL, one must adopt the wave optics theory for the sounding. In the cases where the vertical dimension of atmospheric structures is comparable to the first Fresnel zone of the RO path, wave optics or the hybrid wave-ray methodology must be invoked in analyzing the radio wave propagation process. In a comprehensive monograph, Melbourne [21] described in detail how multi-path propagation and atmosphere-induced spectral broadening can lead to ambiguity in geometric–optic solutions to the GNSS-RO sounding. Figure 3 provides a schematic view of the multi-path problem in the GNSS-RO sounding, where the radio wave propagates through a thin atmospheric layer and is split to waves propagating in the different directions. In essence, this RO split can be seen as an analogy to the single-slit light experiment. As a result, modulated SNR is expected due to self-interference at the RO receiver end, a U-shape feature in the SNR profile. Using a thin-screen model, Melbourne [21] obtained an analytical simulation of the SNR response from a 0.5 km atmospheric layer with a sharp negative $N$ gradient ($\mathrm{d}N/\mathrm{d}z < 0$) at the top and positive gradient ($\mathrm{d}N/\mathrm{d}z > 0$) at the bottom. The U-shape response in the SNR profile is a result of the Fresnel response and the so-called "throw-back" rays from layer diffraction. According to the model simulation, the U-shape gap is related to the layer thickness and can be extracted from the GNSS-RO observation. A similar technique was applied for ionospheric $E_s$ layers to infer the $E_s$ thickness [31].

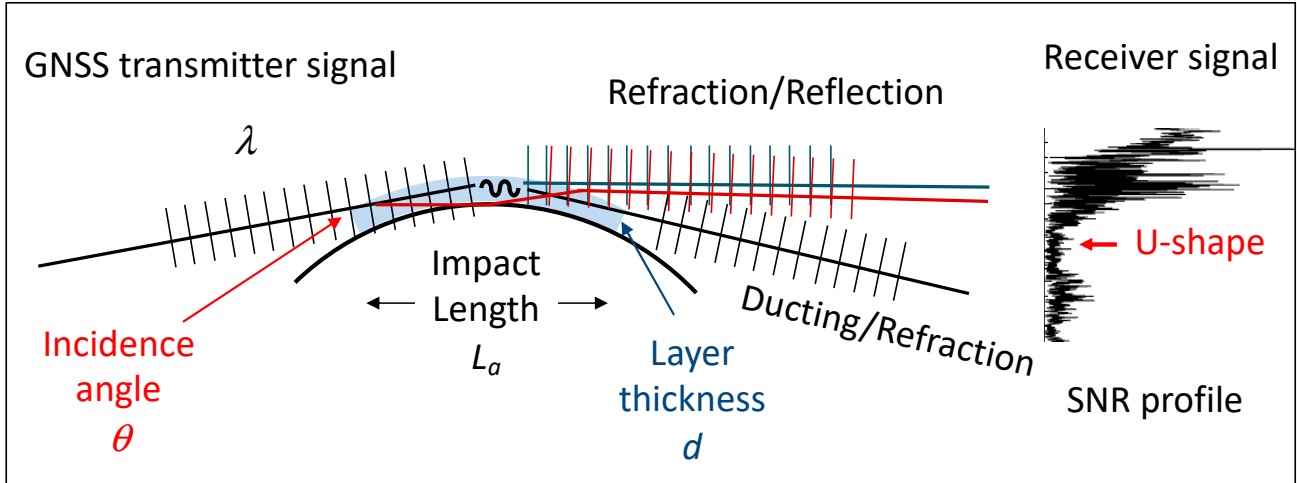

**Figure 3.** Schematic diagram of GNSS-RO wave optics as refracted by an atmospheric layer and reflected by the surface in multi-path interference.

The RO multi-path propagation through a thin atmospheric layer can induce constructive interference, causing re-emerged signals at deep bending angles or $H_{SL} < -130$ km, as seen in Figure 3. A model simulation study showed that the ducting layer with a finite horizontal extension can effectively generate these short-burst deep $H_{SL}$ SNRs [22]. Such ducting layers, either elevated or near the surface, are able to channel the radio propagation through a longer distance and be refracted far into the occulted condition. These simulations showed that the deep $H_{SL}$ $S_{RO}$ is proportional to the horizontal length and vertical gradient of refractivity from the ducting layer.

The deep $H_{SL}$ $S_{RO}$ (below the U-shape) and grazing reflection features (above the U-shape) appear to be related to each other in moist MABL situations, according to these earlier studies [21,22]. The U-shape $S_{RO}$ profile is indicative of the bifurcation generated by a thin ABL layer where the ducting may occur. As expected from a thin-layer model

(Appendix B), the grazing reflection can come from the surface as well as an elevated atmospheric layer. While the reflection features contain coherent oscillations in the $S_{RO}$ profile, the deep $H_{SL}$ signals below the U-shape exhibit little coherence in general. Since ABL thickness is mostly < 2 km, any interference features would be packed as low-frequency power in the $S_{RO}$ power spectrum as suggested by the thin-film model. The deep $H_{SL}$ $S_{RO}$ power spectrum shows mostly red noise with weak and coherent amplitudes. Hence, in this study we simply calculate an average power of the deep $H_{SL}$ $S_{RO}$ and develop an empirical relationship of $S_{RO}$ to MABL $q$.

*2.4. $S_{RO}$ on $H_{SL}$ Coordinate*

One of the disadvantages of working with the GNSS-RO Level-1B data is the lack of information on bending angle and tangent height ($h_t$) that are derived from Level-2 processing. These high-level physical quantities or products require an inversion from the $f_{ex}$ measurements. In the case of low $S_{RO}$ at deep $H_{SL}$, this inversion is impractical because the $f_{ex}$ measurements are too noisy. Thus, the purely geometric variable $H_{SL}$ becomes a legitimate coordinate to register the $S_{RO}$ profile.

As shown in Figure 4, $H_{SL}$ can be different for the RO profile coming from the same bending angle if the LEO satellite altitudes are different. This difference must be taken into account when comparing the $S_{RO}$ measurements from different LEOs. For the satellite altitude variations over a small range, an $H_{SL}$ correction can be made approximately using a scaling factor as below,

$$\hat{H}_{SL} = H_{SL} \cdot [1 + 0.0006 \cdot (812 - H_{SAT})] \qquad \text{for } H_{SL} < 0 \text{ km,} \qquad (4)$$

to shift $H_{SL}$ to a new coordinate $\hat{H}_{SL}$, where $H_{SAT}$ is the LEO satellite altitude with reference at 812 km. Figure 5 shows significant differences between SRO profiles as a function of $H_{SL}$ and $\hat{H}_{SL}$. By applying Equation (4) to four satellites orbiting at different altitudes, Figure 6 shows that the derived monthly climatologies for August are in general agreement with each other after registering $S_{RO}$ on $\hat{H}_{SL}$.

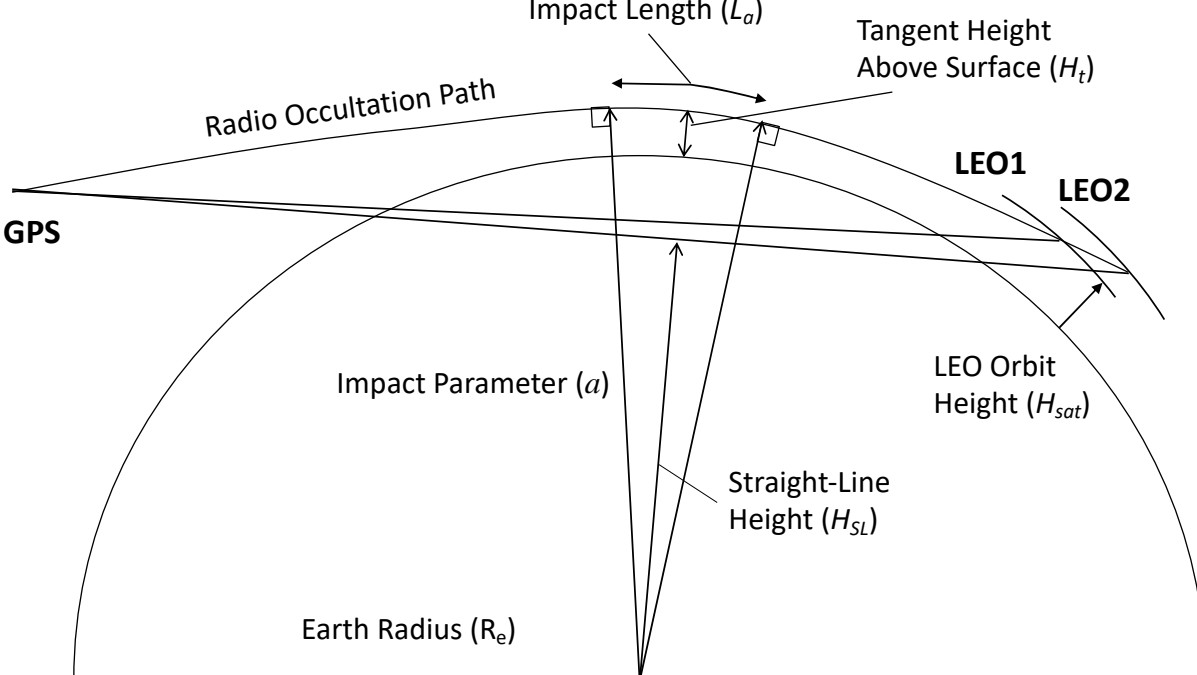

**Figure 4.** Diagram of the RO geometry with two LEO satellites from different orbital altitudes. For the same RO path, the two LEO receivers register the identical bending angle at different $H_{SL}$ heights.

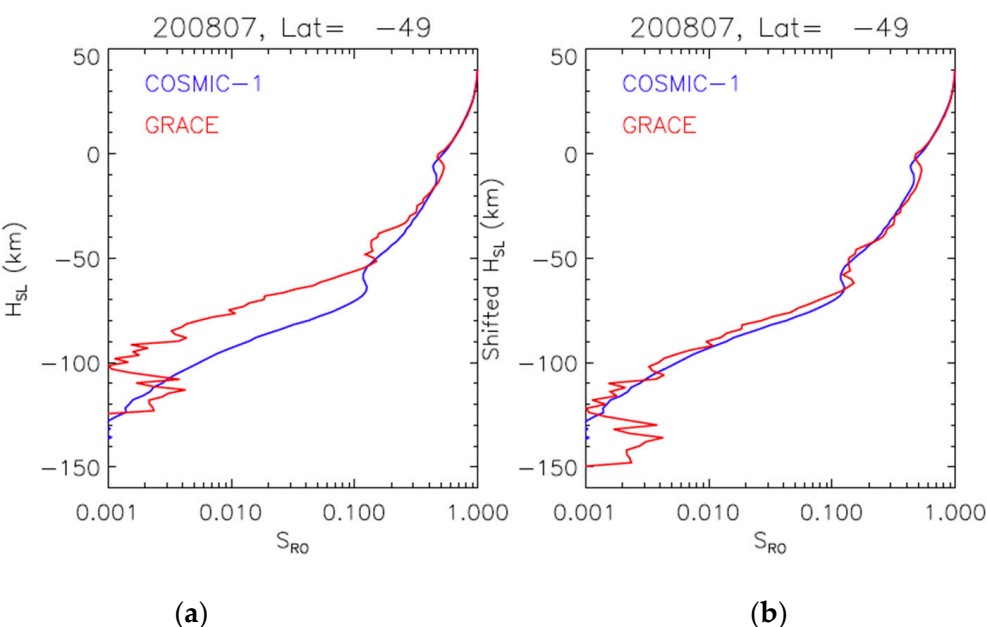

**Figure 5.** The mean $S_{RO}$ profiles from COSMIC-1 and GRACE from July 2008 when their orbital altitudes were respectively at 810 and 350 km: (**a**) as a function of $H_{SL}$, (**b**) as a function of $\hat{H}_{SL}$.

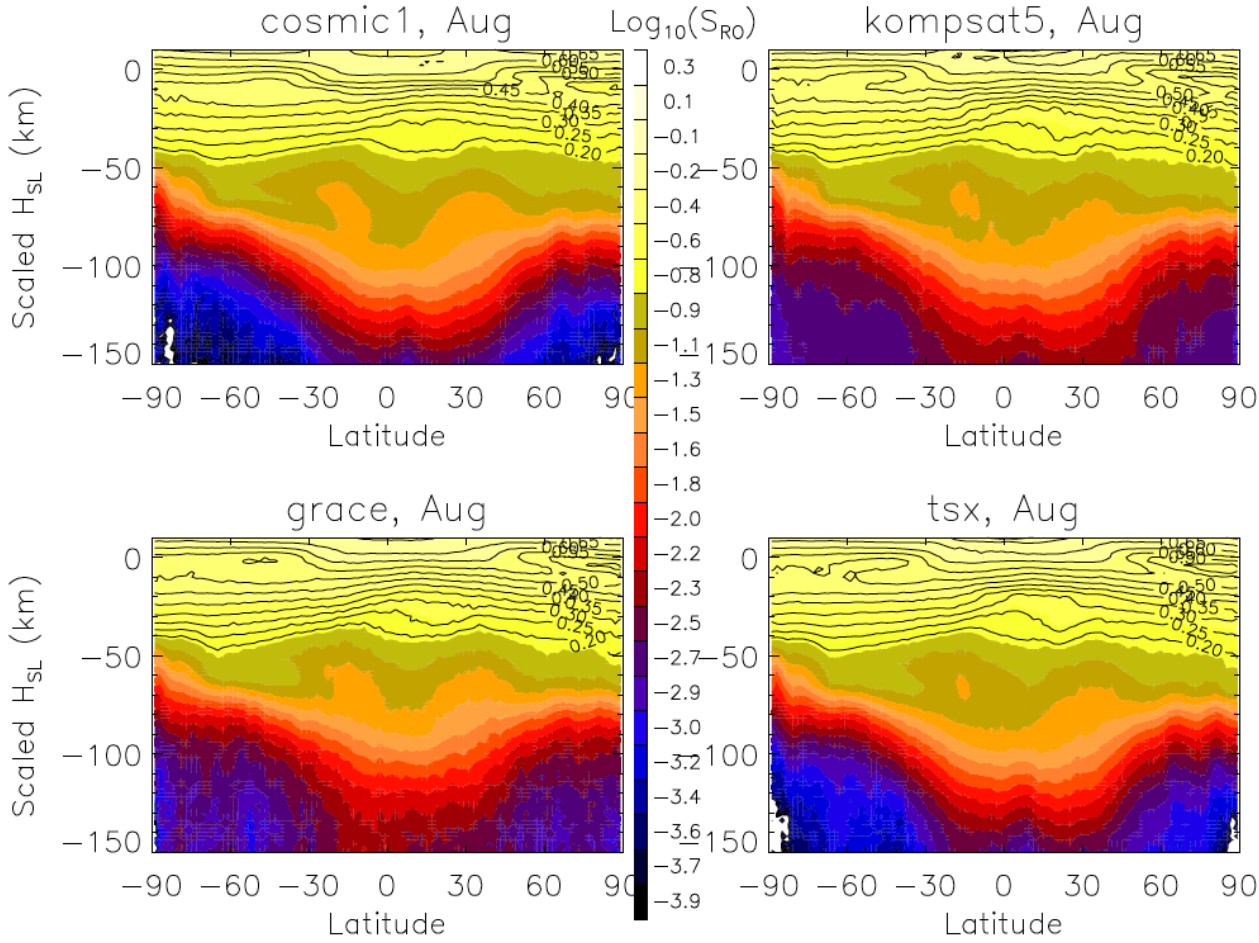

**Figure 6.** Mean August $S_{RO}$ from COSMIC1 (810 km), KOMPSAT5 (560 km), GRACE (300–500 km), and TSX (520 km) as a function of $\hat{H}_{SL}$. The $S_{RO}$ amplitude is colored in a $\text{Log}_{10}$ scale to highlight very small values at deep $\hat{H}_{SL}$, whereas contour lines represent a linear scale of $S_{RO}$.

### 2.5. $S_{RO}$ on $\phi_{exL1}$ Coordinate

An alternative method to register the $S_{RO}$ profile without relying on Level-2 products is to use the L1 excess phase profile ($\phi_{exL1}$). $\phi_{exL1}(h_t)$ is proportional to the atmospheric refractivity $N(h_t)$ at tangent height $h_t$, which is proven to be a good first-order approximation (Appendix C). Since the refractivity is proportional to atmospheric density, this approximation suggests that we may interpret the $S_{RO}$–$\phi_{exL1}$ relation roughly as an $S_{RO}$–pressure relation.

Figure 7 shows the same August $S_{RO}$ climatologies from the four satellites but on the Log10($\phi_{exL1}$) coordinate. Unlike in Figure 6, no empirical scaling is applied to the $\phi_{exL1}$-based vertical coordinate. All $S_{RO}$ and $\phi_{exL1}$ profiles come directly from the Level-1B data. As seen from Figure 7, the $S_{RO}$ on the Log10($\phi_{exL1}$) coordinate shows a consistent climatology among four RO sensors, despite their different satellite orbital altitudes. Hence, the Log10($\phi_{exL1}$) method is considered as a preferred approach when comparing multiple satellite $S_{RO}$ observations.

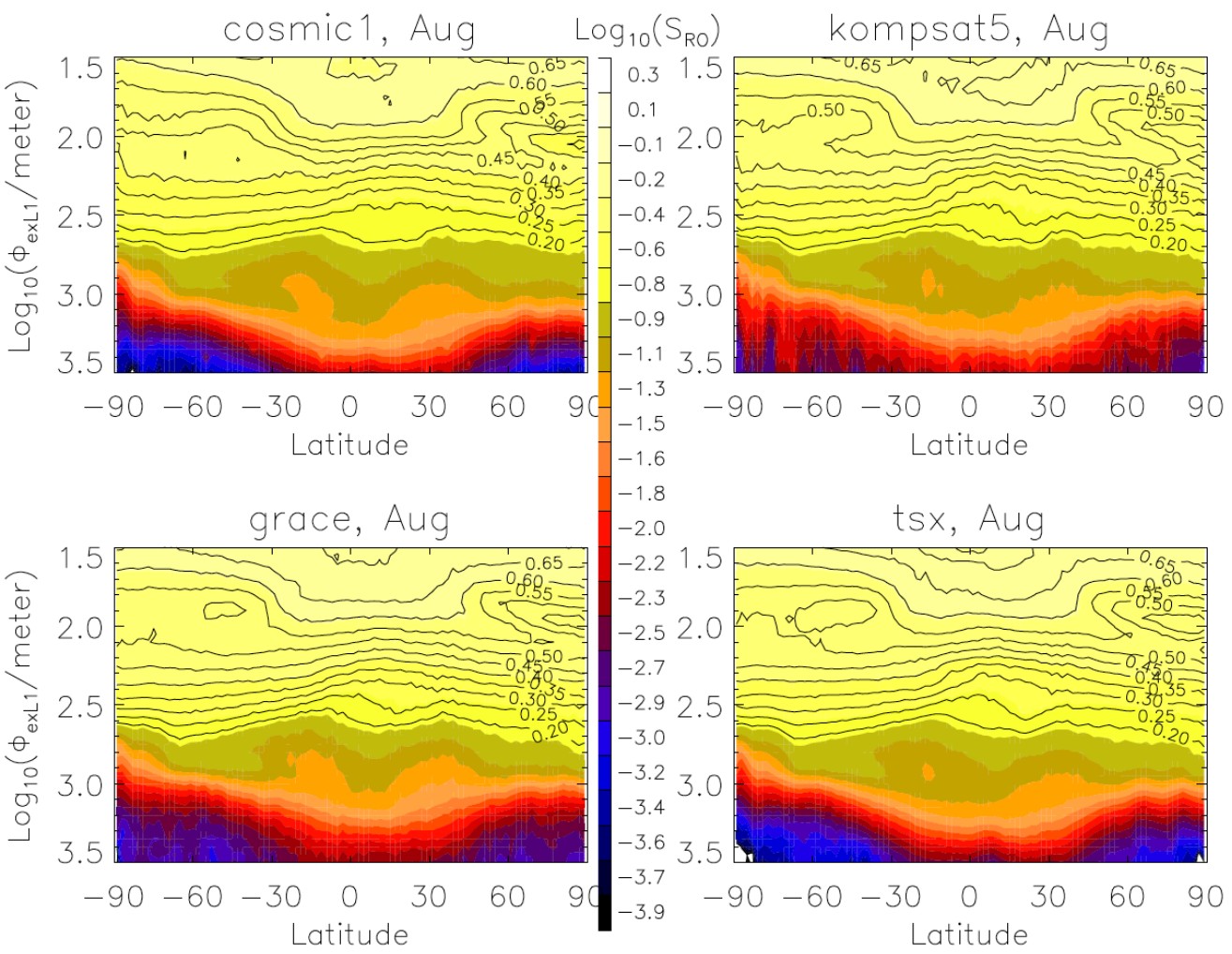

**Figure 7.** As in Figure 6 but for $S_{RO}$ as a function of Log10($\phi_{exL1}$).

### 2.6. Data Quality Control

The $S_{RO}$ data need to be screened for short profiles, failed OL tracking, and very low SNR to ensure data quality for ABL studies. An RO profile is excluded if it produces no data at $H_{SL} = 50$ km or below $-20$ km. The operation CL-to-OL transition often occurs at $H_{SL} = -20$ km, but it can fail in some cases, producing no signal below $-20$ km. These failed profiles create essentially the measurement noise at deep $H_{SL}$ and must be excluded for the data analysis. A threshold method based on $S_{RO}$ mean and standard deviation is developed to identify and exclude the profile with a failed CL-to-OL transition. Finally,

the RO profiles with a very poor signal (SNR < 200) are excluded in this study, since the measurement noise would become too important in these cases.

The deep $H_{SL}$ $S_{RO}$ measurements are sensitive to the jamming of GNSS signals, and these artifacts were not removed in this study. As shown in Appendix D, the jamming of GPS signals has increased substantially in recent years, particularly in the conflict zones, showing large spatial and temporal variations. Although the jamming occurred mostly over land, it produced a significant $S_{RO}$ amplitude in the Mediterranean Sea and some coastal regions.

*2.7. MABL Sampling*

The number of Level-1B $S_{RO}$ measurements after quality screening is still significantly higher than the number of Level-2 products (e.g., temperature, specific humidity) in the MABL. For example, because the number of COSMIC-1 Level-2 products in the MABL is so low, it is impossible to generate a monthly map, let alone to study the diurnal variation of MABL $q$. In addition, for those Level-2 retrievals that reach the MABL, there is likely a bias to the situations where the MABL tops were not too sharp to induce the super-refraction condition.

Figure 8 compares the tropical sampling statistics between Level-2 (COSMIC-1 version 2013.3520 and COSMIC-2 version 0001.0001) and Level-1B (COSMIC-1 atmPhs version 2013.3520 and COSMIC-2 conPhs version 0001.0001) products from the COSMIC Data Analysis and Archive Center (CDAAC) in terms of percentage over the total number of $S_{RO}$ profiles in the tropical (30° S–30° N) MABL. The number of Level-2 atmPrf profiles drops sharply with height from the mid-troposphere to MABL. At 400 m, the number of Level-2 retrievals from January and July 2008 are, respectively, 6.2% and 7.3% of the number of Level-1B $S_{RO}$ profiles that reaches $H_{SL} = -100$ km. The percentages for the global (not shown) are similar to the tropical statistics. The lower number of Level-2 retrievals (i.e., refractivity, water vapor, and pressure) at MABL is largely because it requires $\phi_{exL1}$ measurements at deep $H_{SL}$, which are unavailable where $S_{RO}$ is too noisy or SNR is too low. In the case where there exists a strong vertical $N$ gradient at the top of MABL, the Level-2 algorithm is often unable to carry out the refractivity inversion to the heights below the MABL top.

The statistics of COSMIC-2 retrievals, however, improved significantly, as expected, from the more capable TGRS (TriG GNSS Receiver System) [32,33]. The tropical MABL penetration percentage increased to 71% and 73% in COSMIC-2 atmPrf for January and July 2020, respectively, and to 56% and 59% in COSMIC-2 wetPf2, with respect to the Level-1B sampling at 400 m. The TGRS employs a 6-element antenna array technology for beamsteering and beamforming to improve the gain and SNR of received RO signals, which results in better penetration to the MABL than COSMIC-1 receivers. Together with the matured OL operation, the fast TGRS onboard real-time positioning determination and scheduling also produce more RO profiles per receivers than COSMIC-1. Nevertheless, COSMIC-2 lacks coverage of high latitudes, and therefore in this study we still use the COSMIC-1 data in the following analysis.

In addition to the sampling differences between the Level-2 and Level-1B data, the Level-2 $q$ is known to have a dry bias due to the so-called negative $N$ bias from the Abel inversion. It is found that the $N$ bias cases are often associated with the Abel inversion of the profiles impacted by ducting layers or strong vertical $N$ gradients at the ABL top [16,34]. These biased situations are often found over stratus-topped MABL where the super-refraction and ducting often occur. As indicated in Equation (1), retrieving $q$ or $P_w$ from $N$ requires knowledge about the $T$ profile in the MABL. This auxiliary information usually comes from an atmospheric model or assimilated data. A 1% error in $T$ would lead to a 2% error in the $N$-to-$q$ conversion.

Hence, the $S_{RO}$-based technique offers a promising approach for the MABL $q$ retrieval without requiring a priori information about the atmosphere. The deep $H_{SL}$ $S_{RO}$ profiles do not cut off because of a sharp ABL top, which can produce an unbiased sampling of stratus-

topped clear-sky or cloudy-sky conditions in the MABL. A single-level $S_{RO}$ technique from deep $H_{SL}$ $S_{RO}$, as shown in the next section, has a limited sensitivity range for the MABL $q$. However, this limitation can be largely mitigated by using the $S_{RO}$ data from multiple $H_{SL}$ levels. Adequate characterization of the MABL $q$ variability requires a full diurnal coverage without sampling bias for both clear and cloudy skies, and the $S_{RO}$-based technique has great potential for achieving this goal. Finally, the unbiased sampling is also important for model-observation comparisons and evaluation. As shown in the next section, the MABL $q$ observations over broken clouds and sea ice pose a great challenge for both numerical models, data assimilation, and remote sensing from space.

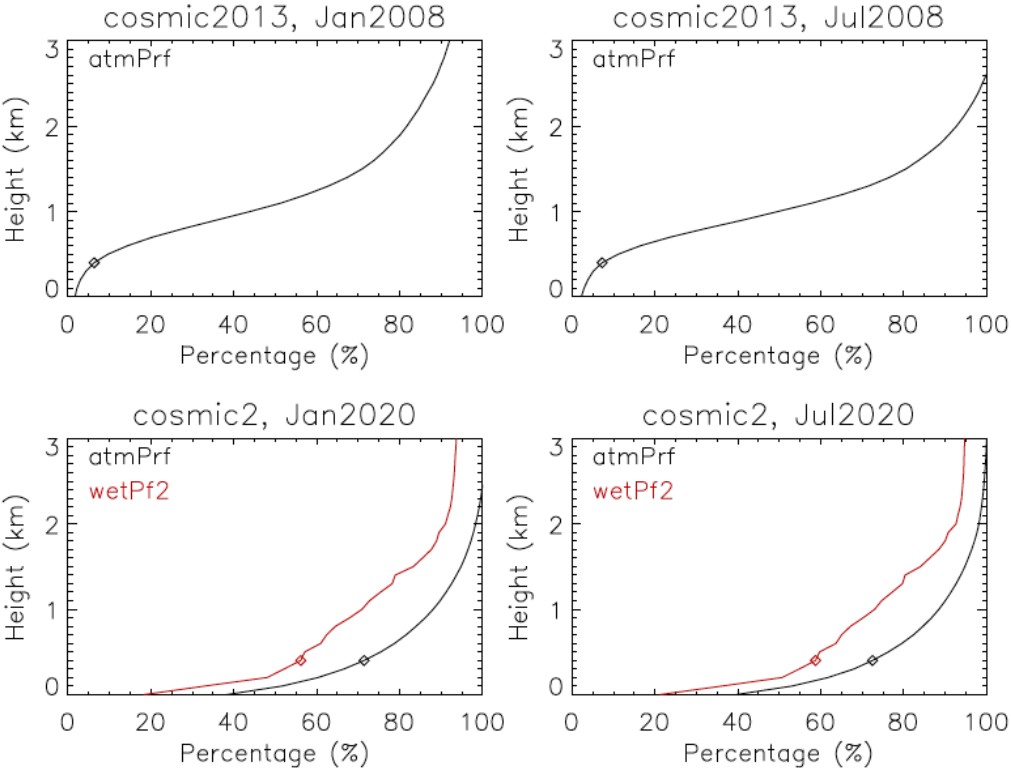

**Figure 8.** Comparisons of tropical (30° S–30° N) MABL penetration percentage of the Level-2 data over the Level-1B $S_{RO}$ from COSMIC-1 (top panel) and COSMIC-2 (bottom panel) in January and July. The 2008 (atmPrf and wetPrf) and 2020 (atmPrf and wetPf2) data are used for COSMIC-1 and COSMIC-2, respectively. Only atmPrf statistics are shown for COSMIC-1 since they are the same as wetPrf. The 400 m level is indicated by the symbol in each panel.

## 3. Results

### 3.1. $S_{RO}$ Sensitivity to MABL $H_2O$

To evaluate the $S_{RO}$ sensitivity to MABL $q$, we compared the monthly mean $S_{RO}$ at a deep $H_{SL}$ from COSMIC-1 with the monthly mean-specific humidity ($q$) at 950 hPa or ~400 m altitude from the European Centre for Medium-Range Weather Forecasts (ECMWF) Reanalysis v5 (ERA5). We selected the $S_{RO}$ data from the single level at $H_{SL} = -100$ km or $Log_{10}(f_{exL1}) = 3.25$ but excluded those from high elevation and rough landmasses in comparison with ERA5 $q$. Figures 9 and 10 show the mean $S_{RO}$ maps from January and July 2008 at $H_{SL} = -100$ km and $Log_{10}(f_{exL1}) = 3.25$ and their comparisons with the ERA5 $q$. There existed a good correlation between the deep $H_{SL}$ $S_{RO}$ amplitudes and the ERA5 $q$ at 950 hPa or 400 m, especially at mid-latitudes where $q$ varied between 3 and 13 g/kg. The correlations of the 950 hPa $q$ with the $S_{RO}$ were similar for both $H_{SL} = -100$ km and $Log_{10}(f_{exL1}) = 3.25$ levels, but perhaps with slightly tighter scatters with the $S_{RO}$ data from $Log_{10}(f_{exL1}) = 3.25$. We evaluated the correlation between the 400 m $q$ and other $S_{RO}$ levels; the $Log_{10}(f_{exL1}) = 3.25$ level exhibited the highest coefficient.

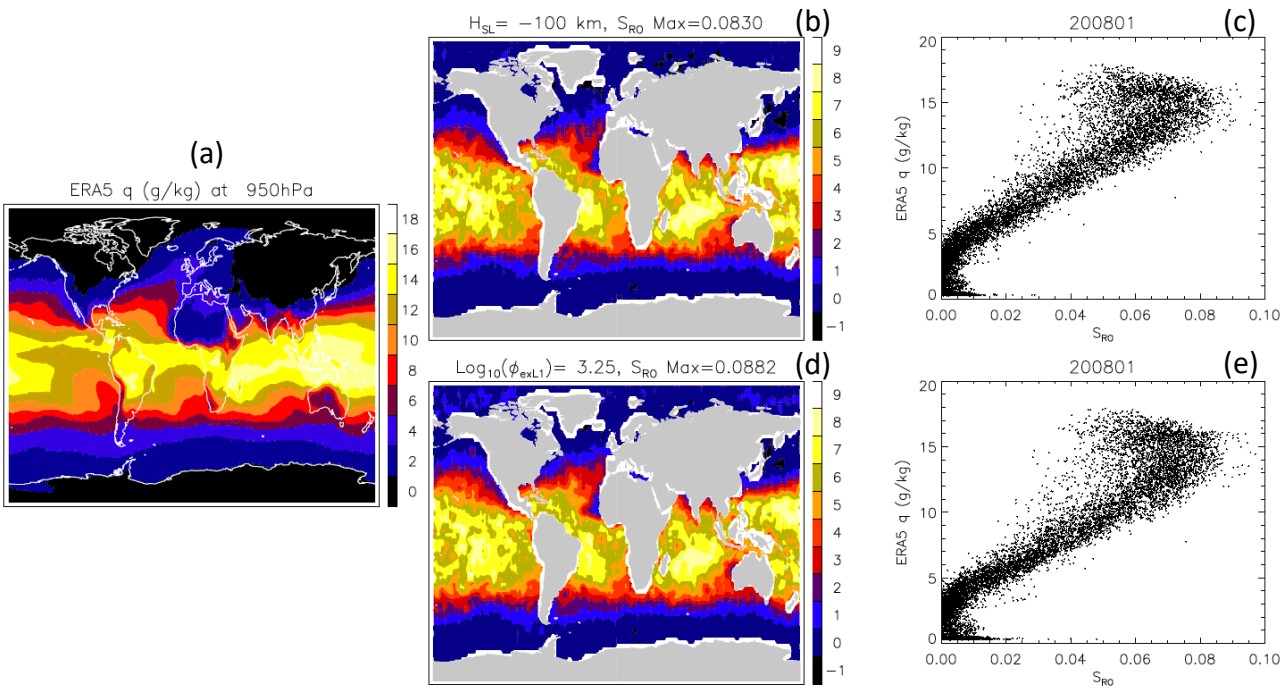

**Figure 9.** Correlation between monthly mean ERA5 $q$ and COSMIC-1 $S_{RO}$ for January 2008: (**a**) ERA5 950-hPa $q$, (**b**) COSMIC-1 $S_{RO}$ at $H_{SL} = -100$ km, (**c**) correction between (**a**,**b**), (**d**) COSMIC-1 $S_{RO}$ at $\text{Log}_{10}(f_{\text{exL1}}) = 3.25$, and (**e**) correlation between (**a**,**d**). Both GNSS-RO and ERA5 data are on a $2° \times 2°$ longitude–latitude grid. The color scales in (**b**,**d**) need to multiply the maximum value in the title to obtain the $S_{RO}$ values on the map. Surfaces with high (>100 m) elevation and large roughness (standard deviation > 30 m) are excluded in this comparison, which are masked in grey.

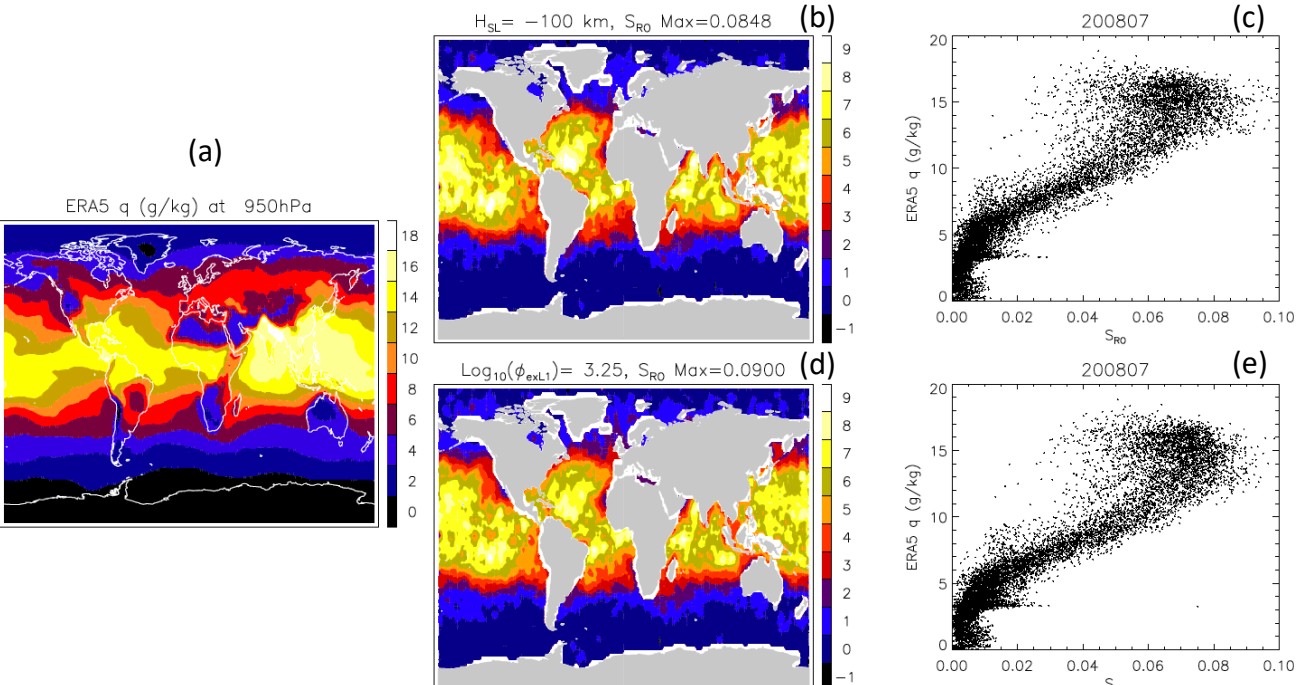

**Figure 10.** As in Figure 9 but for July 2008.

Some notable differences were seen between the $S_{RO}$ and ERA5 $q$ maps in the subtropical MABL, such as the Southeast Pacific (SEP) and the Southwest Atlantic (SWA) in January 2008 (Figure 9) and the Northeast Pacific (NEP) and the Northeast Atlantic (NEA) in July 2008 (Figure 10). These are the MABL regions with frequent occurrence of broken clouds that can be affected by complex processes including surface latent heat flux, enhanced precipitation, downwelling longwave radiative flux, large scale divergence, and aerosol forcing [35]. A comparison of the ERA5 and COSMIC-1 Level-2 $q$ suggested a likely dry bias in the ERA5 data at 3 km [36]. A dry bias of the reanalysis data was also found in the MABL when compared to ship-based radiosonde observations [37]. This bias was on the top of the COSMIC-1 dry bias due to the negative-$N$ bias, as identified in the Abel retrieval method [16,17]. Nevertheless, this was a challenging measurement and required further validation from observations. The $S_{RO}$-based method revealed significant differences from the reanalysis data and COSMIC Level-2 products, which can serve as an independent source of global MABL $q$ measurements for future validation.

In the regions where the MABL $q > 13$ g/kg, the correlation with $S_{RO}$ at $H_{SL} = -100$ km and $Log_{10}(f_{exL1}) = 3.25$ degraded significantly, due to possible saturation by a large CWV in the troposphere. As seen in Figures 9 and 10, these regions were generally associated with the Intertropical Convergence Zone (ITCZ) and the South Pacific Convergence Zone (SPCZ). Although the correlation was poor for this single-level $S_{RO}$, the MABL $q$ was found to correlate with the $S_{RO}$ at other levels (not shown) in a non-linear way. Thus, a new study is ongoing to apply a machine learning and artificial intelligence (ML/AI) method to the MABL $q$ retrieval by utilizing the $S_{RO}$ measurements from all levels. The preliminary results suggest that the ML/AI algorithm is very promising and can retrieve the large $q$ values over the deep convective regions as well as the MABL $q$ at mid-to-high latitudes. The ML/AI MABL $q$ retrievals will be validated against ship-based radiosonde observations from field campaigns.

In the high-latitude dry regions where $q$ was below 3 g/kg, the $S_{RO}$ values from $H_{SL} = -100$ km or $Log_{10}(f_{exL1}) = 3.25$ were too small to establish a meaningful correlation with the ERA5 $q$. However, a good correlation was identified by comparing the July ERA5 950 hPa $q$ with the $S_{RO}$ from $Log_{10}(f_{exL1}) = 3.10$ (approximately, $H_{SL} = -80$ km) for the polar region (Figure 11). As expected, the larger $S_{RO}$ values at $Log_{10}(f_{exL1}) = 3.10$ helped to establish a good $q$–$S_{RO}$ relationship at low MABL $q$ values under the drier condition. Nevertheless, the positive $q$–$S_{RO}$ correlation appeared to be only evident in the Arctic for $q > 3.5$ g/kg. Little correlation was found in the Antarctic, where a large part of the domain is covered by sea ice and the ERA5 MABL $q$ was mostly less than 3 g/kg. There were also significant differences over the Arctic Ocean, where sea ice and open water are mixed during this time of the year. The ERA5 $q$ appeared to have difficulty of reporting values below ~3.5 g/kg and lacked features over the Arctic Ocean. A cluster of the 950 hPa $q$ values was assigned arbitrarily to a fixed value (3.5 g/kg) in ERA5 as a minimum in the Arctic. It is worth noting that the ERA5 $q$ could go below 3.5 g/kg at the midlatitudes outside the polar region where there is no sea ice. These low $q$ values at midlatitudes were positively correlated with the $S_{RO}$ from $Log_{10}(f_{exL1}) = 3.10$, suggesting that the 3.5 g/kg problem could be related to the Arctic surface conditions. In the Antarctic, ERA5 did not produce the enhanced features as seen by $S_{RO}$ over the Weddell Sea and the Ross Sea, which is likely due to the similar challenge over sea ice in the Arctic.

In summary, we found that the deep $H_{SL}$ $S_{RO}$ is useful to infer the MABL $q$, after the GNSS-RO SNR signal is properly normalized. The $S_{RO}$-based method could offer an alternative for global MABL $q$ observations, especially over the challenging domains with broken clouds and sea ice. The $S_{RO}$ measurements showed good correlation with the MABL $q$ at 950 hPa (~400 m), particularly for the subtropical and mid-latitude $q$ with the $S_{RO}$ at $H_{SL} = -100$ km and the high-latitude $q$ with the $S_{RO}$ at $H_{SL} = -80$ km. The deep $H_{SL}$ $S_{RO}$ results support the refractive interference theory for radio wave propagation through a thin moist MABL, which would allow the RO signal to reach the deep $H_{SL}$ from a ducting layer. The $S_{RO}$-based MABL $q$ retrieval has ~1500% (40%) more measurements than the

COSMIC-1 (COSMIC-2) Level-2 products. The MABL $q$ inferred from the stratus-topped subtropics does not seem to have the low bias as seen in the standard Level-2 $q$ products. Improved sampling from the $S_{RO}$-based method is critically needed for studying the MABL $q$ and its large spatiotemporal variability.

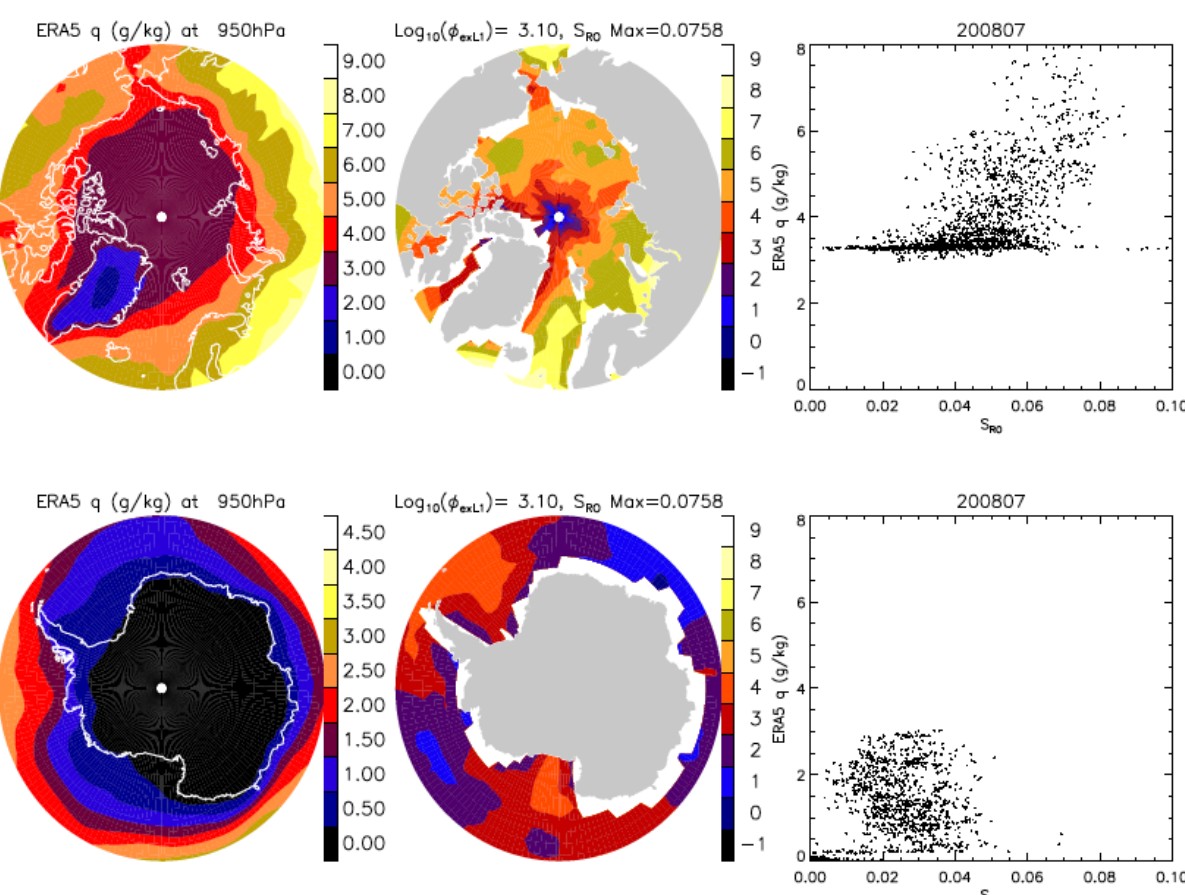

**Figure 11.** As in Figure 9 but for correlation between the 950 hPa ERA5 $q$ and the COSMIC-1 $S_{RO}$ at $Log_{10}(f_{exL1}) = 3.10$ from July 2008 for the Arctic (top panels) and the Antarctic (bottom panels). The data from latitudes of 60° poleward are used in these comparisons.

### 3.2. Diurnal Variations of $S_{RO}$

One of the pronounced tropical and subtropical variabilities is the diurnal variation of MABL $q$. Assuming the MABL $q$ is proportional to the deep $H_{SL}$ $S_{RO}$, we compiled its seasonal statistics as a function of local time for two regions, namely, the Southeast Pacific (SEP) and the Northeast Pacific (NEP), which are defined as (105° W–75° W, 0° S–35° S) and (150° W–120° W, 0° N–35° N), respectively. Because the diurnal variation of MABL $q$ can be a strong function of latitude and longitude, we averaged the $S_{RO}$ data from all the longitudes in each region but kept the local time variation as a function of latitude.

The SEP diurnal variations exhibited very different characteristics from season to season (Figure 12). In June–August (JJA) and September–November (SON), the local time variation was generally larger at subtropical latitudes between 10° S and 25° S. The maximum $S_{RO}$ occurred between noon and early afternoon, with the subtropics lagging slightly behind the tropics. The diurnal cycle was weaker in December–February (DJF) and March–May (MAM), mostly at latitudes between 20° S and 30° S. There was an indication of a semidiurnal variation near 5° S in MAM, which had the highest mean $S_{RO}$ among all seasons. The $S_{RO}$ diurnal variation, as expected for the MABL $q$, was consistent with those reported for cloud amount [38] and cloud liquid water path (LWP) [39] over the same region. A higher cloud amount and LWP were found at night compared to the afternoon.

The higher LWP implies a lower MABL $q$, as seen in Figure 12, because a colder temperature freezes the MABL $q$ into clouds at night. Modeling studies suggested that the cloud and $q$ diurnal cycles in the SEP are mainly driven by the large-scale subsidence originating from the diurnal heating in northern Chile/southern Peru [40]. The observed seasonal variation of the $S_{RO}$ diurnal amplitudes also agreed with the LWP data [39], showing a smaller amplitude in the winter than in the summer.

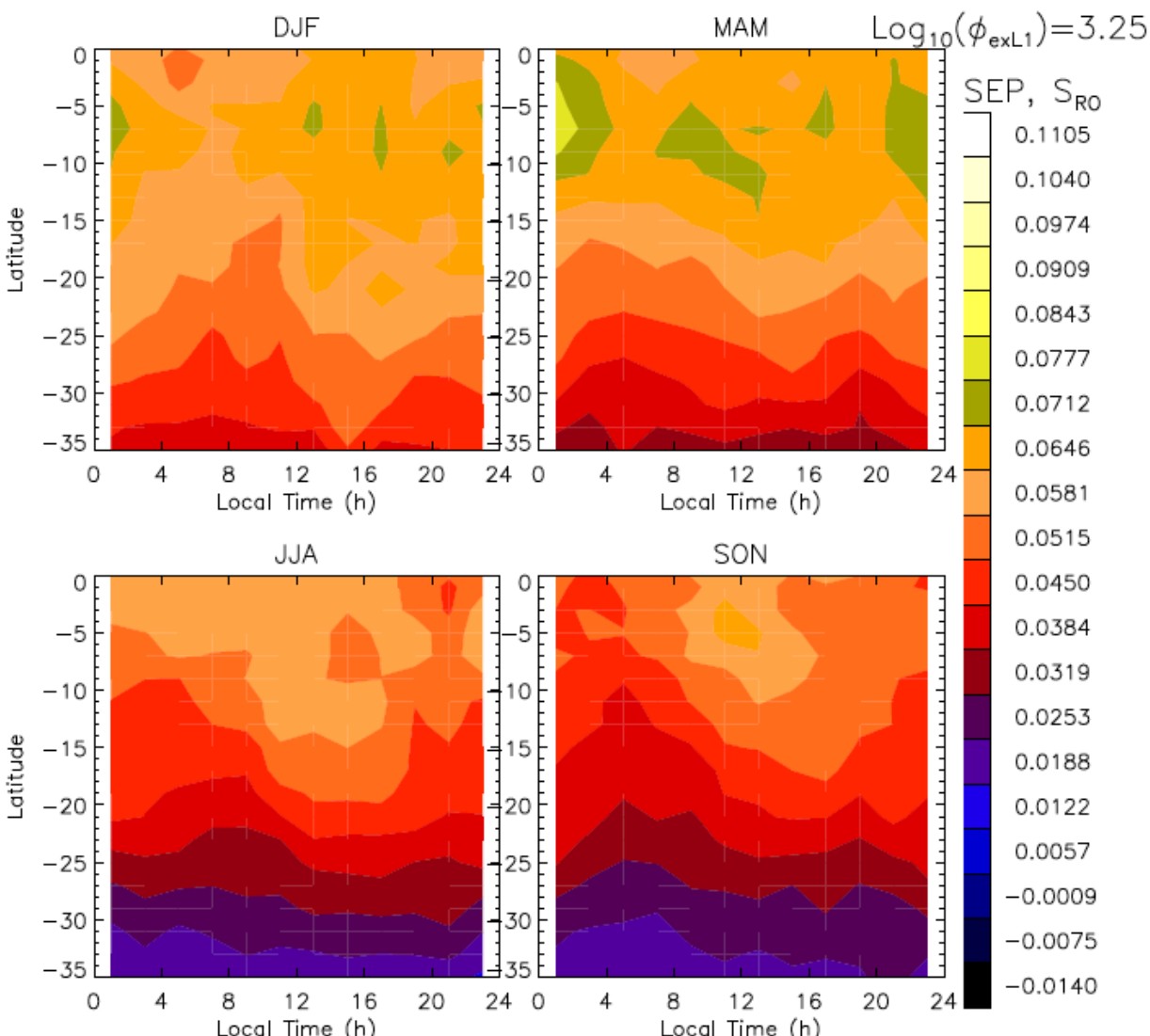

**Figure 12.** Diurnal variations of COSMIC-1 $S_{RO}$ at $Log_{10}(f_{exL1}) = 3.25$ for the SEP region (105° W–75° W). The data from 2008–2013 are averaged into 2-hourly bins to derive the four seasonal climatology at latitudes between 35° S and the equation.

Compared to the SEP region, the $S_{RO}$ diurnal variations in the NEP region were weaker in all seasons (Figure 13), which is consistent with the observed cloud amount variations [41]. The most significant local time variations were perhaps at latitudes between 20° N and 30° N in MAM and JJA, showing an $S_{RO}$ maximum in the afternoon. The low cloud fraction over the NEP was generally lower compared to that in the SEP, which agreed with the higher $S_{RO}$ values in the NEP over the SEP. Given the observed correlation between the $S_{RO}$ amplitude and low cloud fraction variations, it warrants further investigation of this correlation in other regions and under different meteorological conditions.

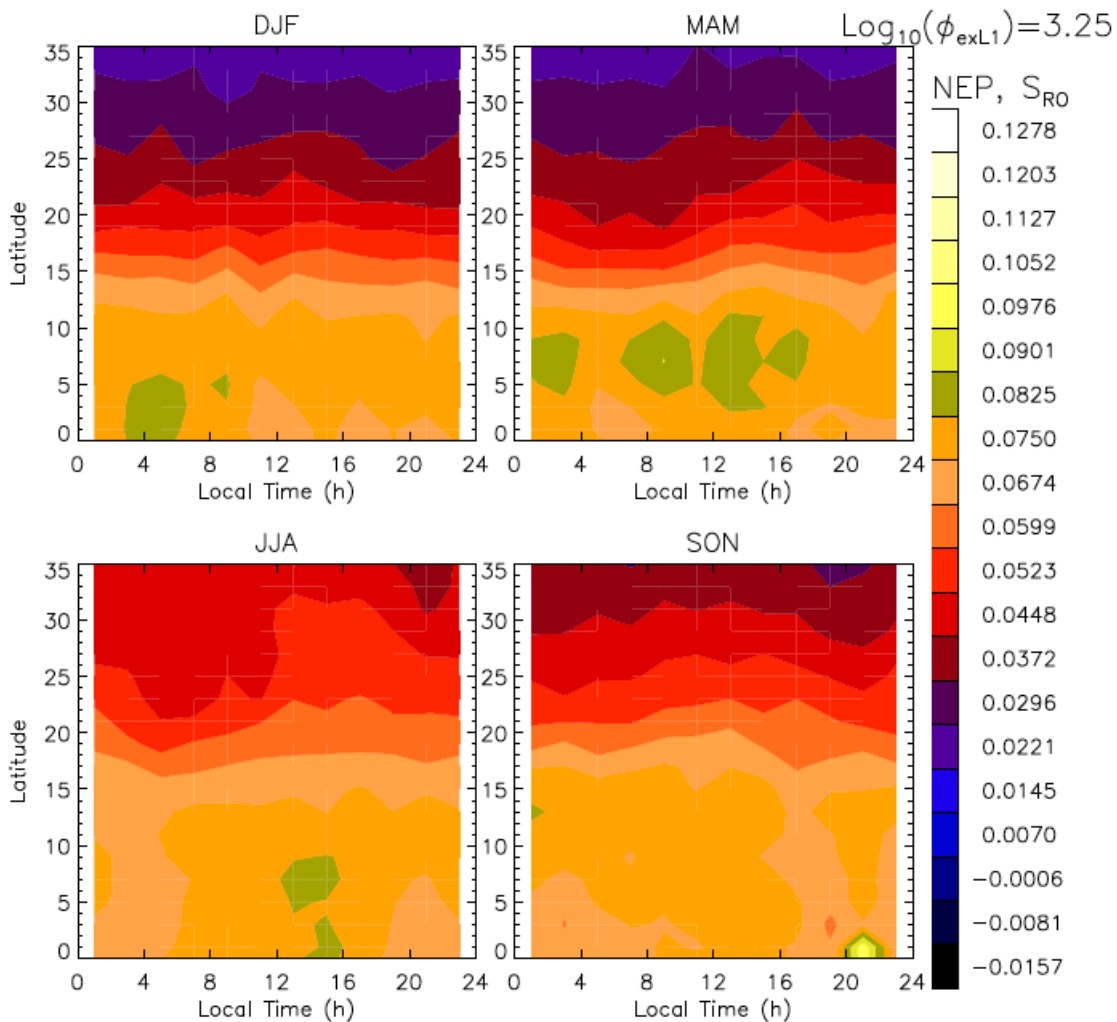

**Figure 13.** As in Figure 12 but for the NEP region (150° W–120° W).

## 4. Conclusions and Future Work

Processes within the MABL play an important role in global cloudiness and require a reliable observational constraint in order for better understanding of their climate sensitivity [42]. Although thermodynamic properties of atmospheric water vapor and temperature are generally available from satellite observations under the Program of Record (POR), lack of vertical resolution, diurnal sampling, and measurement accuracy has been the major obstacle to achieving complete characterization and understanding of the role of MABL in a changing climate.

In this study we presented a promising technique that makes use of the GNSS-RO signal amplitudes at deep $H_{SL}$ to infer the MABL $q$ at 950 hPa (~400 m). The single-level $S_{RO}$ measurements from $H_{SL} = -100$ km show a good correlation with the MABL $q$, especially at the subtropical and midlatitudes, while the $S_{RO}$ from $H_{SL} = -80$ km is useful for the polar region. The $S_{RO}$-based technique offers an alternative approach for the MABL $q$ retrieval without requiring a priori information (e.g., temperature) about the atmosphere. It provides significantly more measurements than the operational Level-2 products, which are needed to cover the full diurnal cycle unbiasedly for clear and cloudy skies, as well as for sharp and smooth ABL tops.

Significant diurnal variations were observed in the deep $H_{SL}$ $S_{RO}$ amplitudes over the Southeast Pacific (SEP) and the Northeast Pacific (NEP) regions. Consistent with low cloud fraction and LWP observations, the GNSS-RO $S_{RO}$ diurnal variations are anticorrelated

with the low cloud amount, as expected if the deep $H_{SL}$ $S_{RO}$ amplitude is proportional to the MABL $q$. Given the good correlation between the $S_{RO}$ amplitude and low cloud fraction variations in the SEP and NEP, further investigation of this relationship in other regions and under different meteorological conditions is warranted.

The inferred MABL $q$ from $S_{RO}$ also revealed significant differences to ERA5 over broken cloud and sea ice conditions. These regions are known to have challenges for numerical models, data assimilation, as well as for remote sensing from space, and they require further validation from other independent MABL $q$ observations. The good $q$–$S_{RO}$ correlation found in this study sheds light on developing a new $S_{RO}$-based MABL $q$ retrieval, especially for these challenging conditions. A subsequent study will apply an ML/AI approach to the MABL $q$ retrieval in which the $S_{RO}$ data from multiple $H_{SL}$ levels are used. It is anticipated that the ML/AI algorithm will be able to overcome the saturation limitation and non-linear dependence of the single-level $q$–$S_{RO}$ relationships.

**Author Contributions:** Conceptualization, D.L.W.; methodology, D.L.W. and J.G.; software, D.L.W.; validation, D.L.W. and M.G.; formal analysis, D.L.W., J.G. and M.G.; investigation, D.L.W., J.G. and M.G.; resources, D.L.W.; data curation, D.L.W.; writing—original draft preparation, D.L.W.; writing—review and editing, D.L.W.; visualization, D.L.W.; supervision, D.L.W.; project administration, D.L.W.; funding acquisition, D.L.W. All authors have read and agreed to the published version of the manuscript.

**Funding:** The work is by supported by NASA's Global Navigation Satellite System Research program under WBS 509496.02.08.13.47.

**Institutional Review Board Statement:** Not applicable.

**Informed Consent Statement:** Not applicable.

**Data Availability Statement:** The COSMIC1 (https://doi.org/10.5065/ZD80-KD74) and COSMIC2 (https://doi.org/10.5065/t353-c093) data used in this study are obtained, respectively, from https://data.cosmic.ucar.edu/gnss-ro/cosmic1/repro2013/ and https://data.cosmic.ucar.edu/gnss-ro/cosmic2/nrt/. The data were last accessed in May 2022.

**Acknowledgments:** UCAR COSMIC Data Analysis and Archive Center (CDAAC) services for data processing and distribution. The authors would like to thank Tae-Kwon Wee for helpful insights and discussions on the COSMIC-1 and COSMIC-2 Level-2 data.

**Conflicts of Interest:** The authors declare no conflict of interest.

## Appendix A. GNSS-RO SNR Variability and Receiver Noise

The GNSS-RO SNR variability is driven by GNSS transmitter power as well as by what part an occultation takes place with respect to the RO antenna field-of-view (FOV). As illustrated in Figure A1, the RO antenna FOV has a pattern to maximize the gain for occultation observations, which directly impacts the RO SNR received. The RO antenna pattern is generally designed to have a wide horizontal but narrow vertical FOV. If the occultation takes place in the middle of the antenna pattern, the signal will be stronger with a relatively higher SNR compared to those that occur on the edge of FOV.

The GNSS-RO receiver noise ($\sigma$) may vary with design, manufacturer, tracking frequency, as well as flight operation. Because this noise is important for the RO radiometry analysis in this study, we needed to determine each receiver's noise individually and remove the noise as expressed in Equation (2). Thus, we developed an empirical method to determine $\sigma$ from the RO at very deep $H_{SL}$ ($H_{SL} < -150$ km) where the majority of RO have no atmospheric signals. As shown in Figures A1–A4, we compiled the statistics of SNR for each receiver from rising and setting ROs and used the mean SNR value at $H_{SL} = -150$ km as the receiver noise. The estimated receiver noise is listed in Table A1.

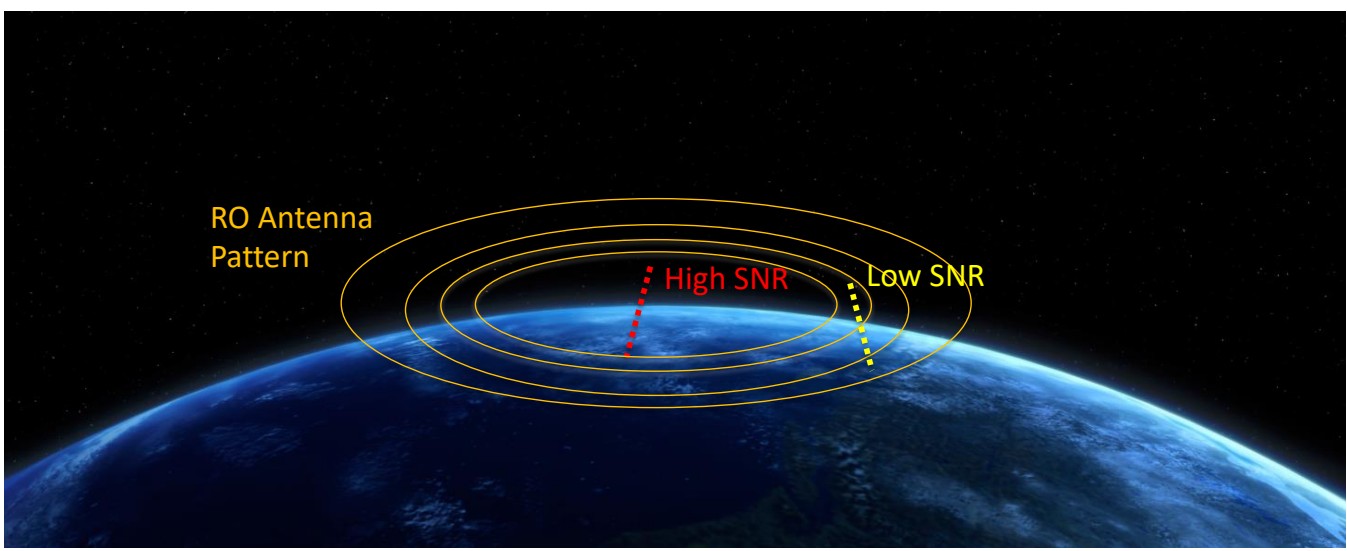

**Figure A1.** GNSS-RO SNR variations with respect to a typical RO antenna pattern. The dotted lines denote occultation events, but their vertical extent is exaggerated slightly to help with illustration.

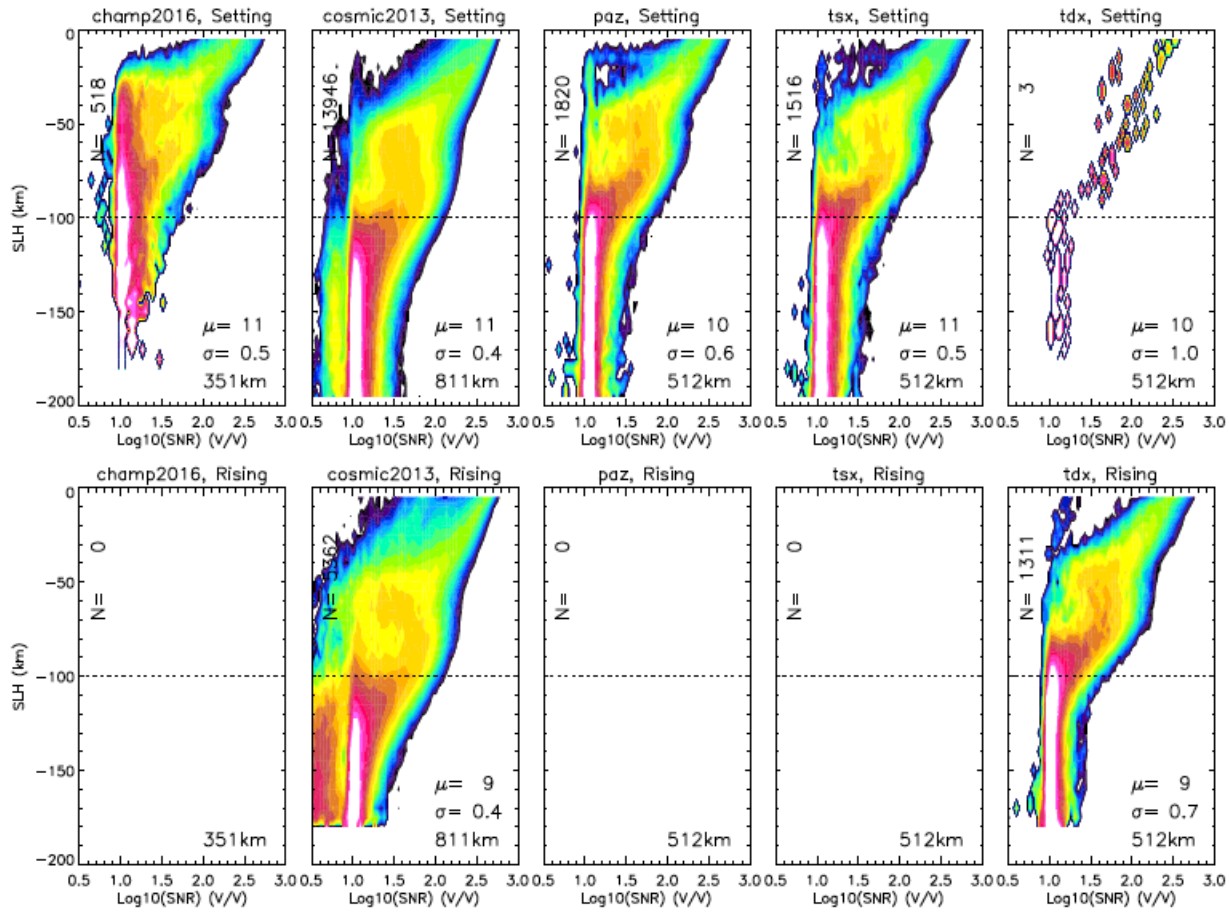

**Figure A2.** SNR statistics for the CHAMP, COSMIC-1, PAZ, TSX, and TDX receivers. The mean SNR values at $H_{SL} = -150$ km are used to determine $m$ (V/V). The 2007 CHAMP, 2009 COSMIC-1, 2020 PAZ, 2019 TSX, and 2019 TDX data are used in the analysis.

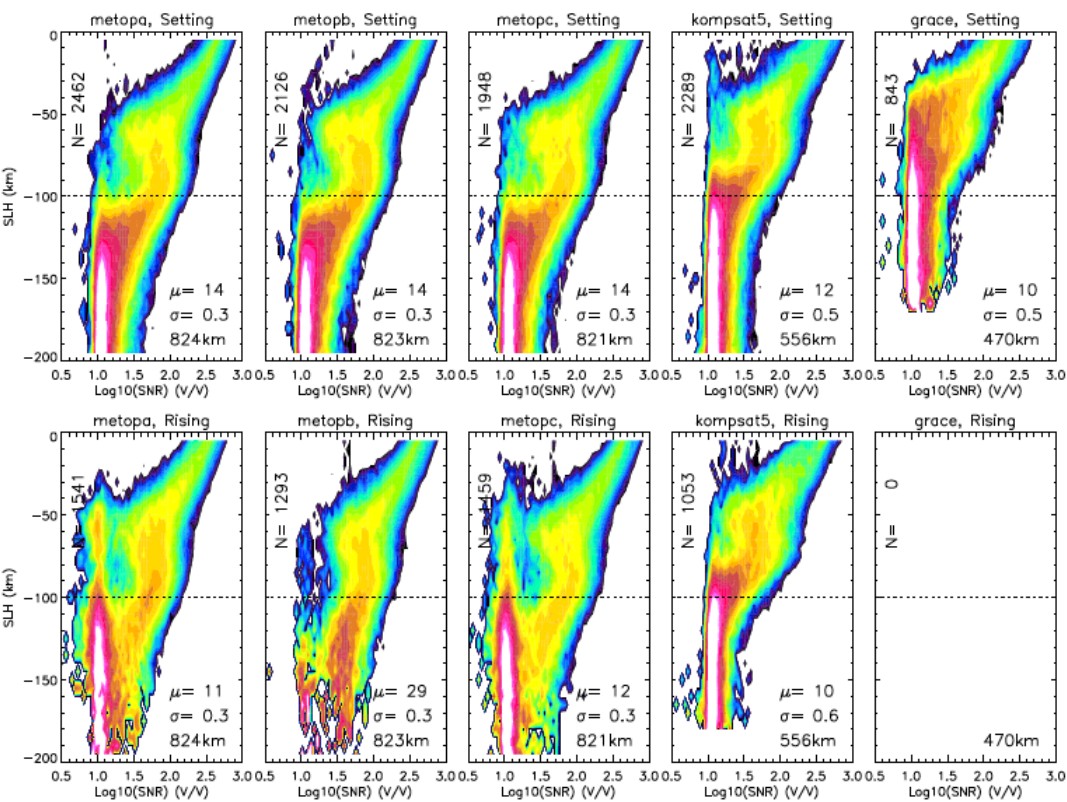

**Figure A3.** As in Figure A2 but for MetOp-A/B/C, Kompsat-5, and GRACE. MetOp, The 2020 MetOp, 2018 Kompsat-5, and 2009 GRACE data are used in the analysis.

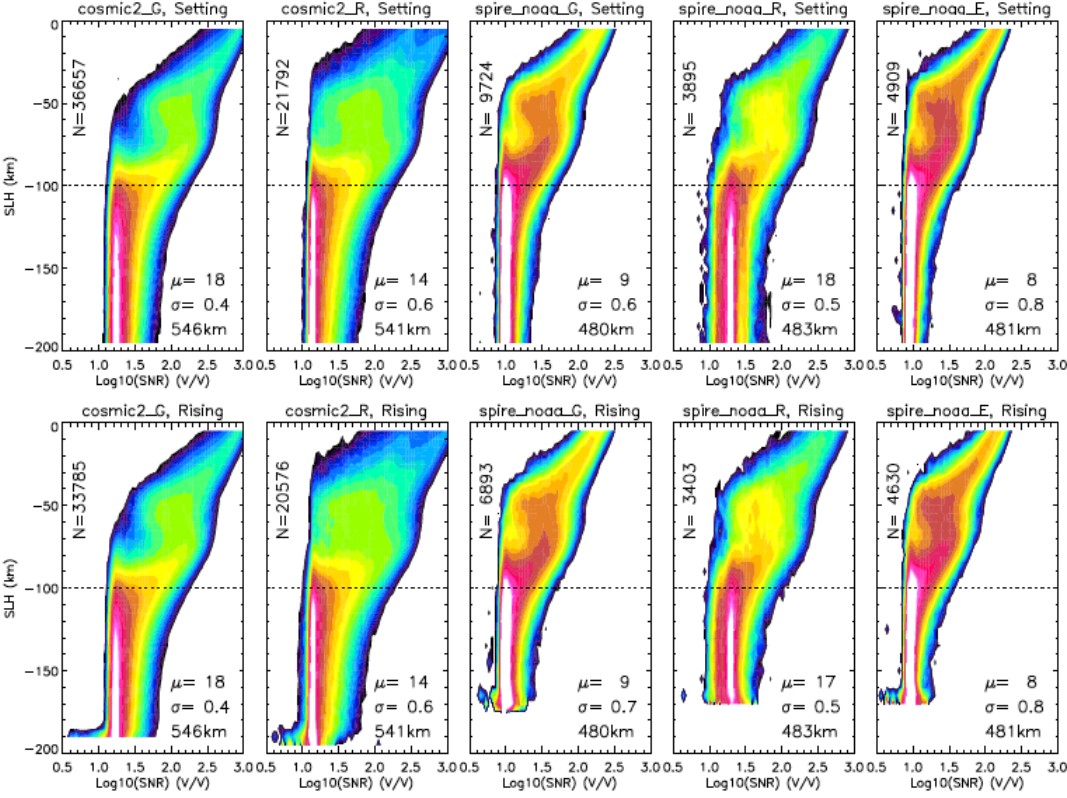

**Figure A4.** As in Figure A2 but for COSMIC-2 and Spire. The 2022 data are used in the analysis.

**Table A1.** Summary of GNSS-RO data in this study.

| LEO Satellites | Mission Lifetime | Init, Final Alt (km) | Sun-Syn (Asc ECT [1]) | Lat Coverage | Top RO Ht (km) | Tracked GNSS | Receiver Noise | |
|---|---|---|---|---|---|---|---|---|
| | | | | | | | Setting | Rising |
| CHAMP | 2001–2008 | 450,330 | varying | 90° S/N | 140 | G | 11 | - |
| COSMIC-1 [2] constellation | 2006–2020 | 525,810 | varying | 90° S/N | 130 | G | 11 | 9 |
| MetOp-A | 2006–2021 | 820 | 19:00 [3] | 90° S/N | 90 | G | 14 | 11 |
| MetOp-B | 2012– | 820 | 19:00 | 90° S/N | 90 | G | 14 | 29 |
| MetOp-C | 2018– | 820 | 19:00 | 90° S/N | 90 | G | 14 | 12 |
| KOMPSAT-5 | 2015– | 560 | 06:00 | 90° S/N | 135 | G | 12 | 10 |
| TSX | 2009– | 520 | 18:00 | 90° S/N | 135 | G | 11 | - |
| TDX | 2016– | 520 | 18:00 | 90° S/N | 135 | G | - | 9 |
| GRACE | 2007–2017 | 475,300 | varying | 90° S/N | 140 | G | 10 | - |
| COSMIC-2 [4] constellation | 2019– | 715,545 | varying | 44° S/N | 90–130 | G | 18 | 18 |
| | | | | | | R | 14 | 14 |
| PAZ | 2018– | 520 | 18:00 | 90° S/N | 135 | G | 10 | - |
| Spire [5] Constellation | 2018– | varying | varying | 90° S/N | 170–400 | G | 9 | 9 |
| | | | | | | R | 18 | 17 |
| | | | | | | E | 8 | 8 |

[1] Ascending-orbit equator crossing time (Asc ECT). [2] The COSMIC1-3 spacecraft never reached the intended orbital altitude and was operated at 725 km for the rest of its mission. [3] The spacecraft started to drift away from the Sun-sync orbit since ~2021. [4] The COMSIC2 NRT data contain GNSS-RO profiles from GPS and GLONASS. [5] The Spire constellation acquires RO profiles from GPS, GLONASS, Galileo, and QZSS (briefly before 2021), and BDS since 2022.

## Appendix B. GNSS-RO Grazing Reflection Features

The RO grazing reflection is a multi-path problem, and the reflection features can be readily seen in the so-called radiohologram [43]. The radiohologram is the power spectrum of RO signals received and sampled by the RO receiver. The radiohologram data are not readily available in the Level-1B products. As an alternative, the running power spectrum profile derived from the $S_{RO}$ profile is used in analyzing the RO reflection features. The running power spectrum can be readily calculated from a detrended Level-1B $S_{RO}$ profile using 1 s running mean differences. The detrended $S_{RO}$ time series is the difference between the original $S_{RO}$ and the 1 s running mean profiles. For the COMSIC-1 (50-Hz) and COSMIC-2 (100-Hz) high-rate RO data, the 1 s running means correspond to the 50- and 100-point average, respectively. As shown in Figure A5, the $S_{RO}$ power spectrum profiles can capture the reflection features reasonably well, similar to those reported by the radiohologram technique. In fact, the radiohologram and the $S_{RO}$ power spectrum profile are closely related in the RO measurements. The RO SNR is the amplitude of the radiohologram at $f = 0$. Since each RO profile reported at the same high sampling rate (50 or 100 Hz), the RO signal at $f \neq 0$ will be sampled elsewhere in the SNR profile at a slightly different time. Therefore, the power spectrum from the detrended $S_{RO}$ profile can capture the reflection features at $f \neq 0$ frequencies. However, unlike the radiohologram, the $S_{RO}$ power spectrum cannot determine whether the RO signal is centered at the tracking frequency.

Leveraging the concepts developed by some earlier studies [21,22], we further examine the branch of SNR distribution above the U-shape. To associate the grazing reflection to the interference patterns seen in the $S_{RO}$ spectral power profile (Figure A5), we apply the thin-film model to this problem:

$$2n \cdot d \cdot \cos(\theta) = (m + 1/2)\lambda \tag{A1}$$

where the interference of wave propagation is defined by elevation angle $q$, layer thickness $d$, radio wavelength $l$, fringe order of interference $m$, and refractivity index $n$. In this simplified model, $n \sim 1$ and the elevation angle $q$ can be related to the bending angle $a$ on the observing platform. As the occultation progresses, $a$ is sampling through different fringe numbers $m$.

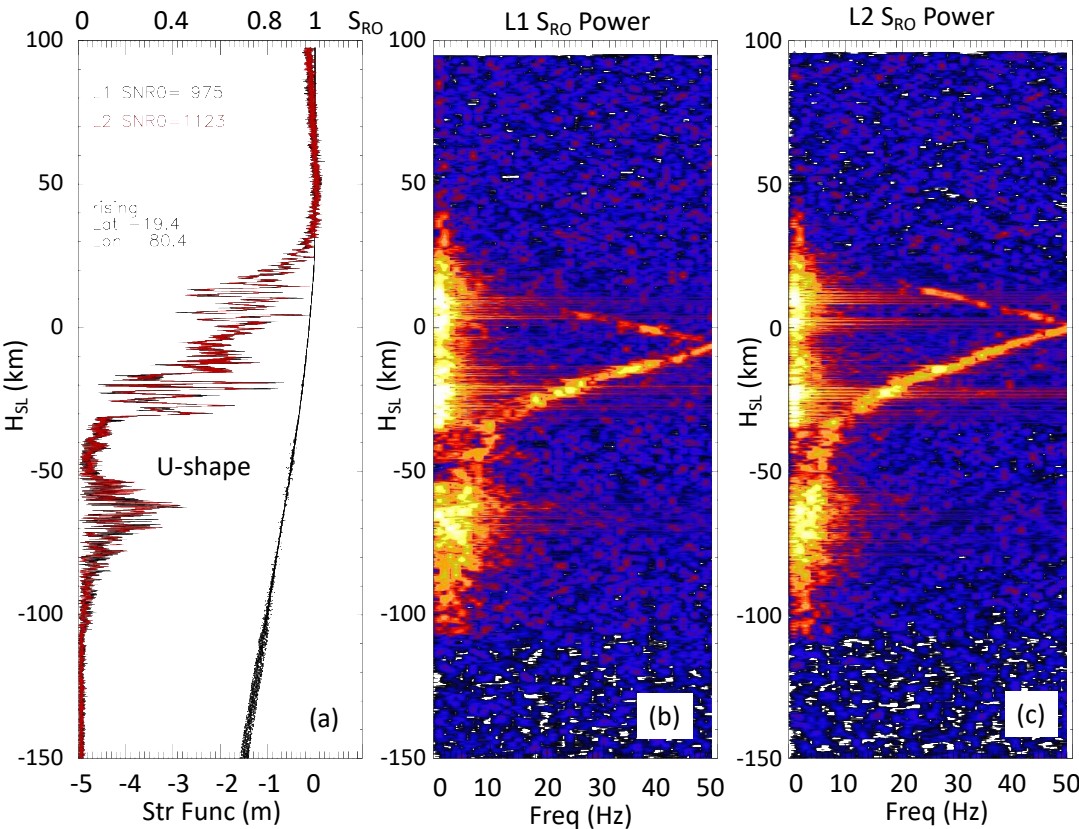

**Figure A5.** (**a**) An example of COSMIC-2 rising occultation at (19.4° S, 80.4° E) with $S_{RO}$ profiles from L1 and L2 channels and structure function of $f_{exL1}$; and (**b**,**c**) L1 and L2 $S_{RO}$ power spectrum profiles, respectively, with the spectral power colored in a logarithmic scale. The free-space $SNR_0$ of L1 and L2 signals are 975 and 1123 V/V, respectively. The structure function is defined as the difference of $f_{exL1}$ measurements between adjacent $H_{SL}$.

The thin-film model in Equation (A1) can explain most of the features as seen from the GNSS-RO grazing reflection. Figure A6a,b show how the direct and reflected rays from a single layer are approximated at elevation angle $q$. The thin-film model can be generalized to multi-layer reflection, as shown in Figure A6c, to allow the reflected RO signals coming from multiple surfaces. RO signals simultaneously reflected from Earth's surface and elevated atmospheric layers are possible. As described by Melbourne [21], a sharp refractivity vertical gradient (e.g., water vapor, temperature) can act as a reflecting surface to produce a partially reflected signal with the same polarization. Thus, it is possible for some of the RO signal to be reflected by an elevated atmospheric layer and some penetrating down and reflected by the surface, to produce multi-layer reflection features. The thin-film model also helps to understand the ducting case Figure A6d from the simulation [21], showing that a very strong SNR can be generated from the interferences out of a ducting layer. As expected from the thin-film model, the ducting over a long impact length would generate a larger SNR and perhaps a wider U-shape SNR void.

Figure A7 shows the interference patterns calculated from the thin-film model (Equation (A1)) when applied to the $S_{RO}$ power spectrum profile as sampled by the RO at a setting rate of 1.3 km/s. The model suggests that the $S_{RO}$ interference patterns are likely induced by a thick atmospheric layer from low fringe numbers. Interferences from layers

with thickness < 5 km and high fringe number would result in the $S_{RO}$ spectral power resided mostly in lower frequencies. For a given layer, the thin-film model reveals that the interference from the L2 frequency would occur at a higher elevation angle than the L1 (Figure A7b), because of the wavelength-dependent fringe number. The fringe–frequency curve in Figure A7, as predicted from Equation (A1), is generally consistent with the observed L1 and L2 reflection features in Figure A5b,c, showing that the L2 pattern is shifted to a slightly higher $H_{SL}$ (i.e., elevation angle).

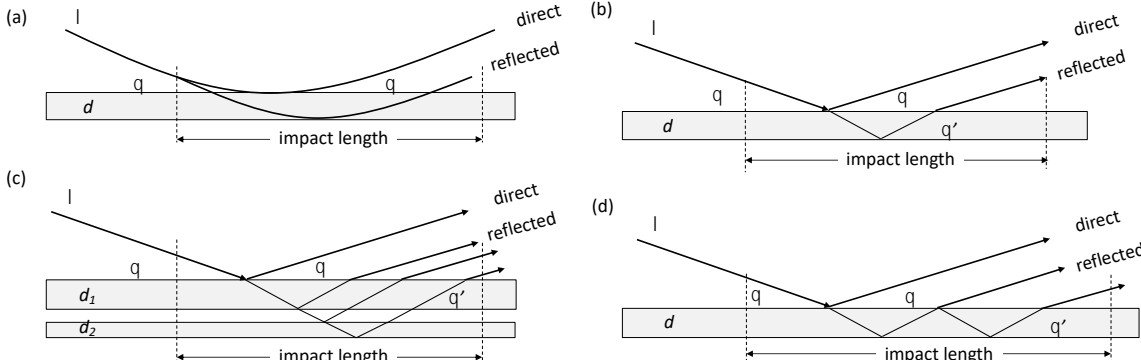

**Figure A6.** (**a**) RO limb sounding rays in a flat surface projection; (**b**) RO direct and single-layer reflected paths in a thin film model; (**c**) RO direct and double-layer reflected paths in a thin film model; and (**d**) RO ducting in a thin film model. The impact length is the region where RO experiences a significant bending as defined in Figure 1, and the elevation angle (*q*) increases as $H_{SL}$ decreases.

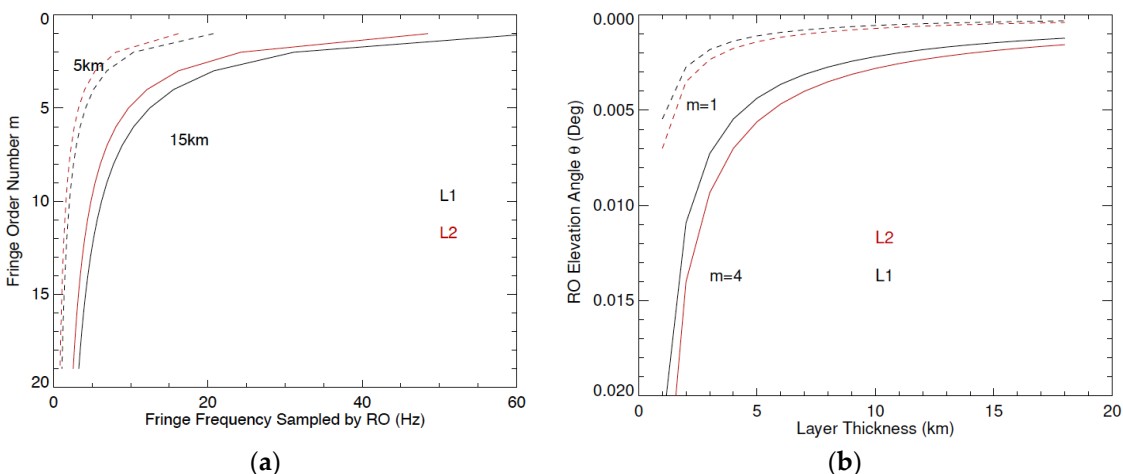

**Figure A7.** RO grazing angle interference patterns predicted from a single-layer thin-film model: (**a**) Fringe number *m* as a function of RO sampling frequency; (**b**) RO elevation angle *q* as a function of layer thickness *d*. A RO setting rate of 1.3 km/s is used for the calculations in (**a**), where two thicknesses (5 and 15 km) are illustrated. Two fringe numbers (*m* = 1 and *m* = 4) are illustrated in (**b**).

Although most of the observed reflection features appear as the single-layer phenomenon, multi-layer features have also been observed in the $S_{RO}$ power spectrum. Figure A8 is an example of double-layer interference, where two branches of the interference feature were found in the COSMIC-2 L1 and L2 $S_{RO}$ power spectra with a weaker amplitude in the upper branch. Because the COSMIC-2 receivers have a 100 Hz sampling rate, they can only resolve the interference features up to 50 Hz. The interference features at a frequency higher than the Nyquist limit would be aliased to the lower frequencies and appear as a folded feature, as seen in Figure A8. As expected from the thin-film model, the L2 interference features occur at a higher $H_{SL}$ or elevation angle than those in the L1. The lower fringe numbers are associated with a lower frequency.

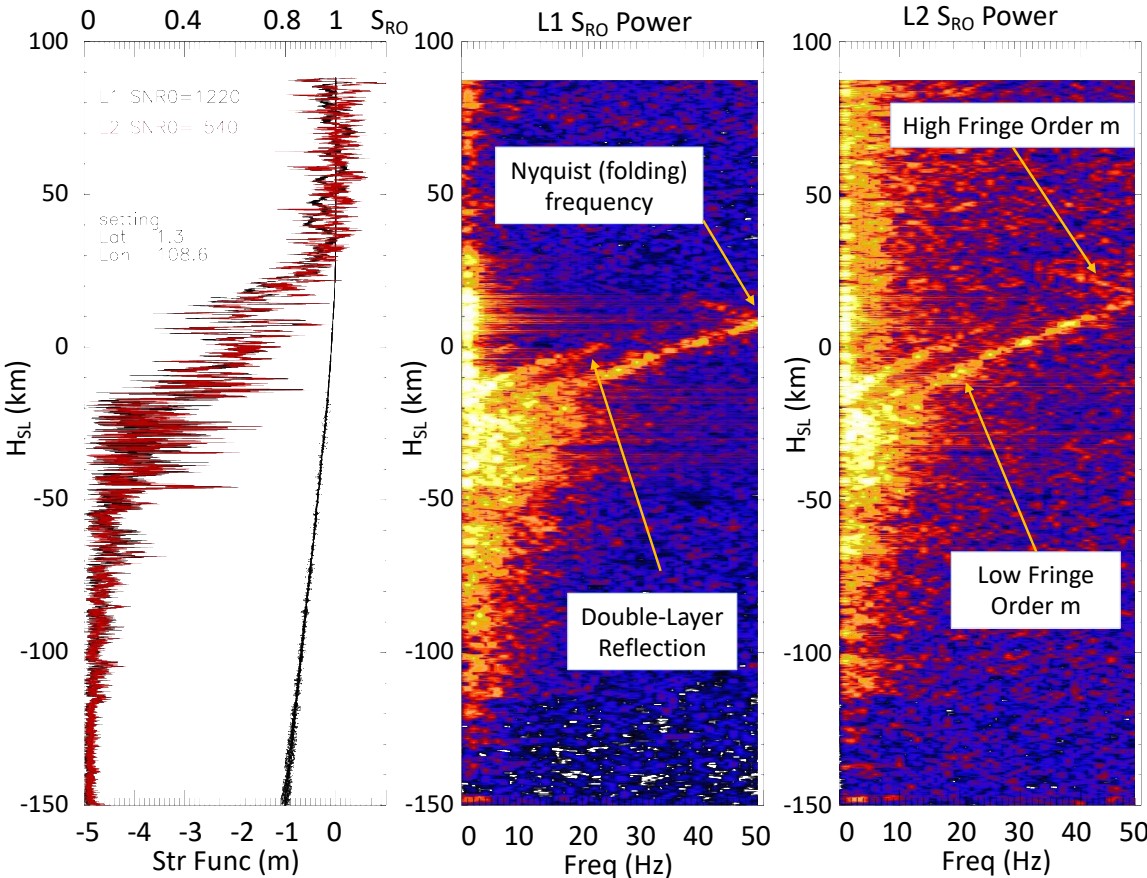

**Figure A8.** As in Figure A5 but for double-layer reflection from a COSMIC-2 grazing angle RO on 11 April 2021 at (1.3° N, 103.6° E), where the reflection feature is evident in both L1 and L2 $S_{RO}$ power spectrum profiles.

Various techniques have been developed to manually/automatically identify and extract grazing reflection features from the RO radiohologram [24,43–45]. In this appendix we developed a new method that allows detection of the reflection features from the $S_{RO}$ power spectrum and provide a more quantitative score of these features. This method can be readily applied to the Level-1B data for both SNR and phase measurements. In addition to the score assigned to each reflection feature, the algorithm can also provide the height of the reflection occurrence in $H_{SL}$.

Figure A9 detailed the reflection detection method and application to the $S_{RO}$ measurements. Because most of the reflection features occur between $S_{RO} = 0.1$ and $S_{RO} = 0.9$, the algorithm is focused on the region of $H_{SL}$ between −60 and 20 km. Figure A9b shows an $S_{RO}$ profile from the COSMIC-2 100 Hz sampling where a reflection feature is clearly evident in the high-frequency $S_{RO}$ power spectrum. Since the $S_{RO}$ spectral power can vary exponentially between low and high frequencies, we only used the digitized $S_{RO}$ power spectrum (Figure A9d). For the $S_{RO}$ power density greater than a threshold ($10^{-6}$ Hz), it is assigned to +1, and otherwise to −1. In addition, because the low-frequency $S_{RO}$ power density contains little information on reflection features, as shown in Figure A9d, this part of the spectrum is excluded in the analysis.

The digitized $S_{RO}$ power density is integrated along five different slopes as indicated in Figure A9d to produce a bit sum for feature identification. This integration is scanned through all $H_{SL}$ levels to produce a bit-sum profile for each slope (Figure A9c). The highest peak from these slopes yields a preliminary score for the reflection feature as well as the $H_{SL}$ location where the reflection occurs. To better separate between weak reflection features on a noisy power spectrum, we divided the preliminary score by a standard

deviation of the digitized $S_{RO}$ power spectrum in the domain to produce the final score for the reflection feature. The five pre-selected slopes used in this were arbitrary, mainly for a computationally efficiency consideration. More slopes may be incorporated in the algorithm to improve detection of the reflection features with different slopes in the $S_{RO}$ power spectrum profile.

A monthly climatology of RO reflection features from COSMIC-1 is shown in Figure A10. In terms of mean score (i.e., reflection strength), frequency of occurrence, and average reflection $H_{SL}$, as derived by the new algorithm. The monthly mean score is a simply averaged score produced by the algorithm for individual profiles. The frequency of occurrence is for the reflection cases with a score higher than 1.6, which generally contains a very clear reflection feature as seen in Figure A10. For the mean reflection height, the cases with a score greater 0.8 are included, which allows more measurements to be used for mapping out this noisy parameter. It becomes quite clear in the score maps that GNSS-RO measurements have fewer grazing reflection features over landmasses and in the moisture tropics and subtropics. These reflection characteristics are generally consistent between the score and occurrence frequency maps. In the polar regions, sea ice and Antarctic and Greenland ice sheets are more reflective in winter than in summer. At the low-to-middle latitudes, atmospheric CWV appears to be the major determining factor of the observed reflection features, as suggested by the higher score and frequency of occurrence in the drier regions. The CWV impact revealed in Figure A10 is in general agreement with the early report on the grazing reflection occurrence derived from a different technique [24].

The $H_{SL}$ height of the RO reflection feature is an interesting property from the new algorithm. As seen in Figure A9, an RO grazing reflection, although sometimes weak, can come from an elevated atmospheric layer above the surface. The mean reflection height maps in Figure A10 suggest that the reflection surfaces are low in the polar regions, especially over sea ice, whereas they are higher in the tropics, as expected from atmospheric moisture layers in the mid-troposphere. It is worth noting that the new reflection detection algorithm is not limited to identify the single reflection layer. As described in Figure A9, it can record multiple peaks if needed to report multi-layer reflection features.

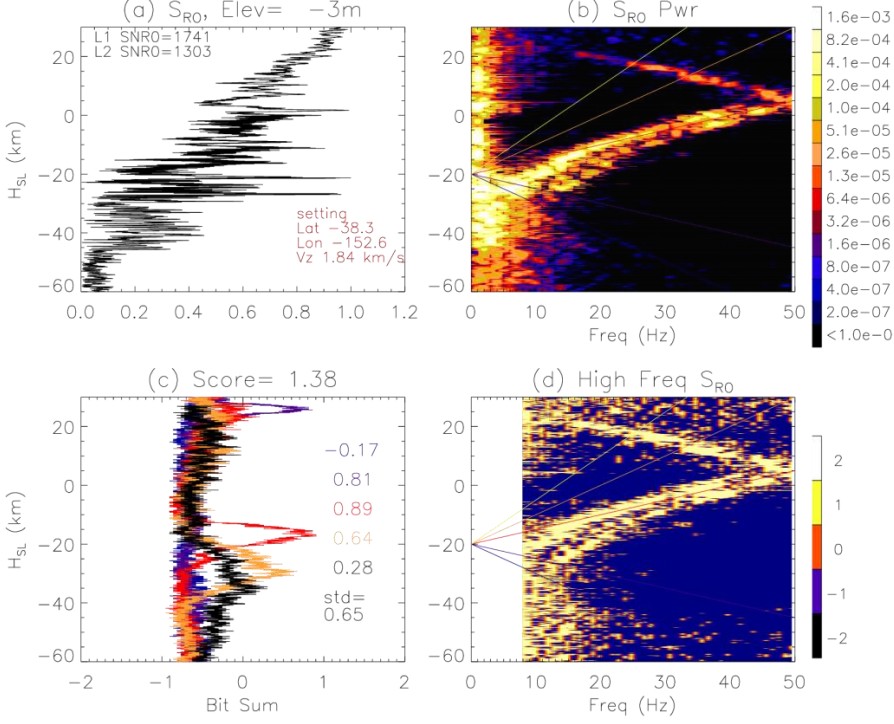

**Figure A9.** Detection and score methods for reflection features in the RO SNR at $H_{SL}$ between −60 and 30 km.

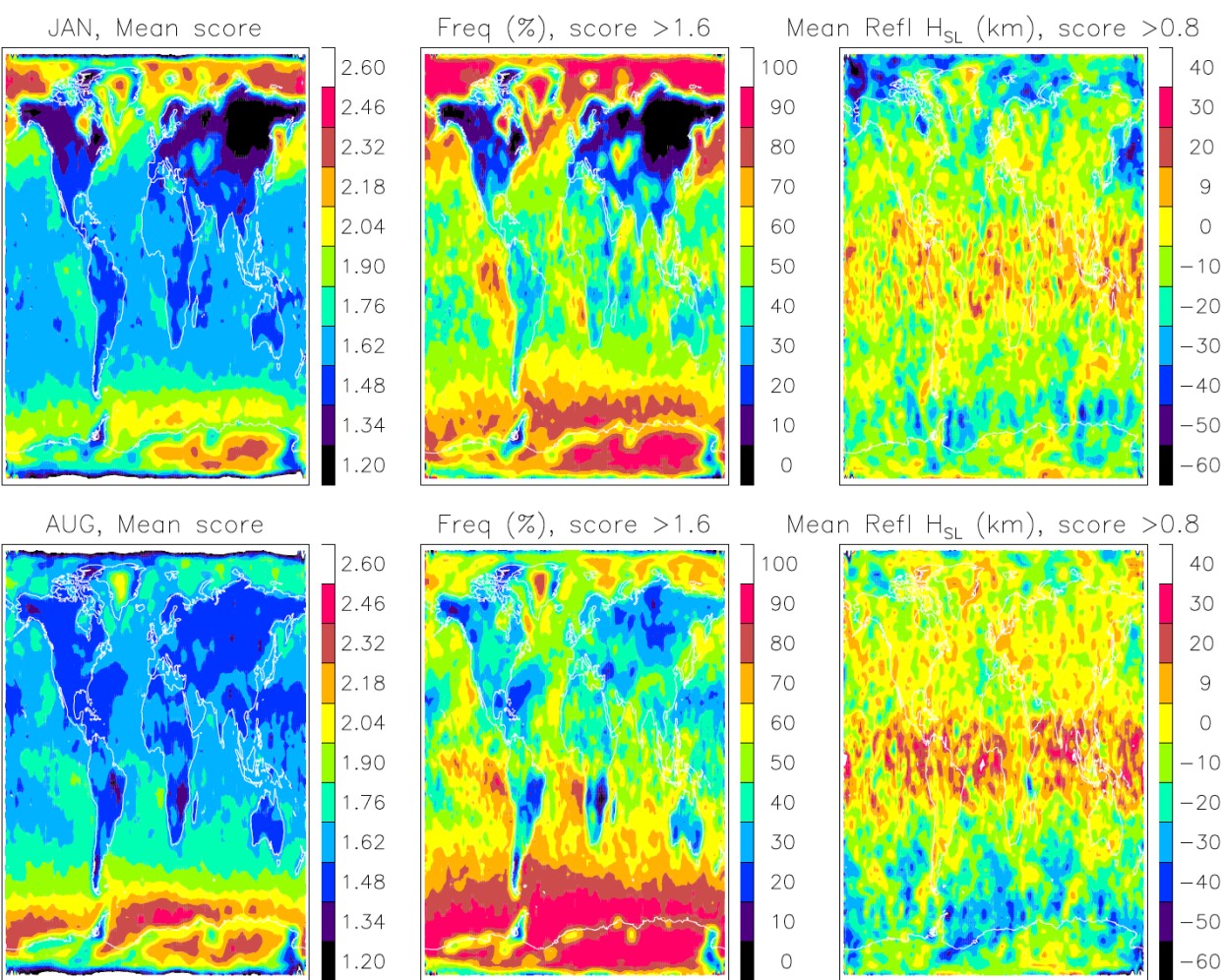

**Figure A10.** Maps of mean reflection score and occurrence frequency for January and August. Three years (2008, 2009, 2010) of the COSMIC-1 data and a 2° × 2° latitude–longitude grid are used to compile the global statistics.

**Appendix C. Excess Phase, Bending Angle, and Refractivity**

GNSS-RO excess phase, bending angle, and refractivity profiles all show an exponential increase in the lower atmosphere. This section describes the first-order relationships between these variables, assuming that atmospheric density decreases exponentially with height.

In geometric optics approximation, the bending angle of radio waves through a spherically symmetric atmosphere can be related to the index of refraction as,

$$\alpha(h_t) = -\int_{r_1}^{\infty} \frac{2h_t}{n\sqrt{n^2 r^2 - h_t^2}} \cdot \frac{dn}{dr} dr \tag{A2}$$

where $a(h_t)$ is the bending angle at tangent height $h_t$, and $r$ is radius. The refractive index $n$ is related to refractivity $N$ by,

$$n = 1 + N \times 10^{-6} \tag{A3}$$

By introducing $z = nr$, Equation (A1) is reduced to,

$$\alpha(a) = -\int_{h_t}^{\infty} \frac{2 \times 10^{-6} h_t}{\sqrt{z^2 - h_t^2}} \cdot \frac{dn}{dz} dz \tag{A4}$$

If we assume the atmosphere refractivity profile has exponential dependence on height, i.e., $N(z) = N_0 e^{-(z-R_e)/H}$, Equation (A3) can be rewritten as,

$$\alpha(a) = -\int_{h_t}^{\infty} \frac{2 \times 10^{-6} h_t}{\sqrt{z^2 - h_t^2}} \cdot \frac{N_a}{H} e^{-(z-h_t)/H} dz \tag{A5}$$

where $H$ is scale height, and $N_t$ is the value of $N$ at $r = h_t$. By substituting $x = z - h_t$, we can reorganize Equation (A4) to,

$$\alpha(a) = \frac{2 \times 10^{-6} h_t N_t}{H} \cdot \int_0^{\infty} \frac{1}{\sqrt{(2+x)x}} e^{-(h_t/H)x} dx \tag{A6}$$

For $h_t/H \gg 1$, which is valid for radio occultation, the integral is largely determined by the range of $x \ll 1$. Thus, we may use the approximation $\sqrt{(2+x)x} \approx \sqrt{2x}$. and reduce Equation (A5) to a gamma function

$$\alpha(h_t) \approx \frac{2 \times 10^{-6} h_t N_t}{H} \Gamma(0.5) \sqrt{\frac{H}{2h_t}} \approx 2.51 \times 10^{-6} N_t \sqrt{\frac{h_t}{H}} \tag{A7}$$

The RO bending angle can be related to the excess phase through a simplified wave propagation model. Melbourne [21] obtained a first-order linear relation between bending angle $a$ and the excess Doppler (i.e., excess phase derivative with respect to time):

$$f_D = V_\perp \sin \alpha \tag{A8}$$

where $f_D = \frac{1}{2\pi} \frac{d\phi_{\text{exL1}}}{dt}$ is the excess Doppler in Hz, and $V_\perp$ is the motion of the LEO satellite perpendicular to the RO line of sight (LOS). For a rising/setting occultation, $V_\perp$ is approximately the ascending/descending rate of tangent height $h_t$ or impact parameter $a$ (i.e., $V_\perp \cong \frac{dh_t}{dt}$). For small bending angles, $\sin \alpha \approx \alpha$, we have,

$$\frac{d\phi_{\text{exL1}}}{dh_t} \approx 2\pi\alpha \tag{A9}$$

From Equations (A6)–(A8) and $N(z) = N_0 e^{-(z-R_e)/H}$, we have,

$$\phi_{\text{exL1}} = \frac{2\pi \cdot 2.51 \times 10^{-6} N_0}{\sqrt{H}} \int_{h_t}^{\infty} e^{-(z-R_e)/H} \sqrt{z} dz \tag{A10}$$

Again, assuming $h_t/H \gg 1$ and $h_t \approx R_e$, we have

$$\phi_{\text{exL1}}(h_t) \approx 1.8 \times 10^{-5} N_t \sqrt{H \cdot R_e} \tag{A11}$$

where $\phi_{\text{exL1}}$, $H$, and $R_e$ all have units of meters. In summary, to the first order, the excess phase $\phi_{\text{exL1}}(h_t)$ is proportional to the atmospheric refractivity at tangent height $h_t$.

**Appendix D. GNSS Signal Jamming**

Jamming on GNSS signals can have a significant impact on the SNR-based remote sensing since it is essentially a radiometry technique. Jamming also poses a great risk to civilian air traffic, regional police, and medical emergency operations. Due to the increased military use of GNSS-navigated weapon systems, jamming from electronic warfare has become common in conflict zones, especially to GPS signals [46,47].

Figure A11 compares the impacts of jamming on GPS and GLONASS $S_{RO}$ signals in Europe and Africa from two selected periods (February 2020 and July 2021). Because the $S_{RO}$ from deep $H_{SL}$ is sensitive to weak GNSS-RO signals, small enhancements from the GNSS jamming can be readily detected on the top of natural atmospheric refractive

signals. Although most of the jamming occurred over land, the ship-based jamming has also been found often in news reports. For February 2020, significant jamming on GPS can be seen in Turkey, Syria, Bulgaria, and Somalia; for July 2021 strong GPS jamming is found in Azerbaijan, Turkey, Mediterranean Sea, Tunisia, and South Sudan. Unlike GPS, the jamming on GLONASS signals was much weaker in these periods. Compared to dramatically different distributions and amplitudes from the jamming, those $S_{RO}$ from natural variability over oceans are consistent between GPS and GLONASS observations.

The time series of $S_{RO}$ signals from two frequently jammed regions shows that GPS jamming has increased substantially since 2017 compared to GLONASS (Figure A12). The increasing jamming on GPS signals is evident in all $S_{RO}$ measurements, despite different receiver types, operational LEO satellite altitudes, and local time samplings, while the jamming increase on GLONASS signals is relatively small from COSMIC-2 observations. As revealed in Figure A11, the jamming power was much stronger in the Mediterranean Sea and Middle East (MSME) region compared to the Central Africa (CA) region, suggesting more presence of electronic warfare in the MME region. A spike of the $S_{RO}$ jamming signals is evident in the data from MetOp-A/B/C and COSMIC-2 for a short period of the 2020 summer over the MSME region. Generally speaking, the daytime jamming amplitude of $S_{RO}$ is larger than that of the nighttime. As also revealed in Figure A12, the GPS jamming in the MME region, which started in 2017, perhaps occurred slightly earlier those in the CA region (rising in ~2018).

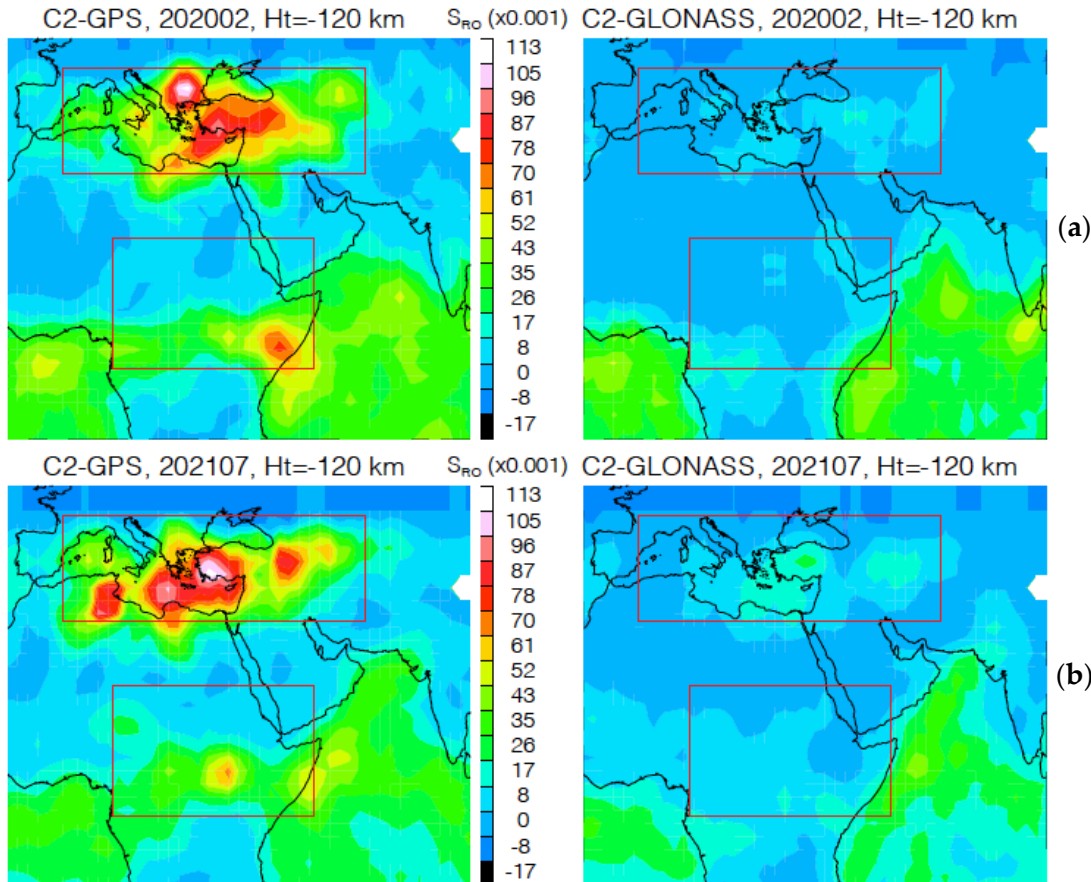

**Figure A11.** Comparisons of GPS and GLONASS jamming signals as observed in the COSMIC-2 nighttime $S_{RO}$ at $H_{SL}$= −120 km over the Mediterranean Sea and Middle East (MSME) and Central Africa (CA) from (**a**) February 2020 and (**b**) July 2021. The MME and CA regions are indicated by the red boxes on the maps.

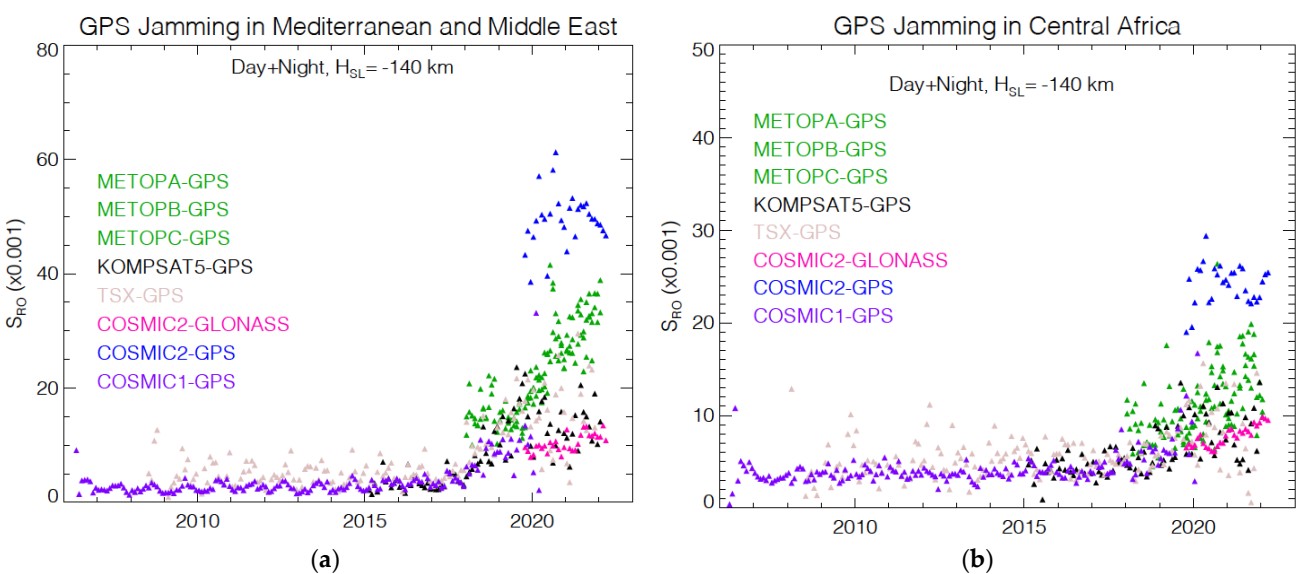

**Figure A12.** Time series of the monthly averaged GNSS jamming (**a**,**b**) amplitude as observed in the $S_{RO}$ at $H_{SL} = -140$ km. The noisier data from TSX and KOMPAST5 than others are mostly due to a fewer number of RO measurements from these satellites. The earlier period of MetOp-A/B $S_{RO}$ data are excluded in the plot because of their testing periods for the open-loop operation.

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
