# Peer review of "GNSS-RO Deep Refraction Signals from Moist Marine Atmospheric Boundary Layer (MABL)"

_atmosphere, doi:10.3390/atmos13060953_

Round 1

Reviewer 1 Report

In this work, tjhe authors proposed a method based on radio occultation signal amplitudes at deep HSL to infer the Moist Marine Atmospheric Boundary Layer at 950 hPa (~400 m). According to the authors, their results  suggest that the deep-HSL SRO is useful to infer the MABL q, after the GNSS-RO SNR signal is properly normalized. This approach may offer an alternative for global MABL q observations, especially over the challenging domains with broken clouds and sea ice. I'm not an atmospheric scientist but I know RO methods pretty well. From the point of view of RO application and measurements, the paper seems very consistent and the processing is fairly described. I consider that the paper may be accepted for publication after very adjustments. Congratulations to the authors for their work.

Remarks:

- The brief mentions about ML/AI in the manuscript seem unnecessary and do not contribute to the work in general. In the abstract, for example, it is mandatory that this mention be removed since this cannot be the contribution of this work. In the conclusion the mention as future work is acceptable.

-a reference about the OL operation is necessary in section 2

-line 101: Melbourne is not a monographa and the reference number ir wrong

Reviewer 2 Report

This paper,  presents a statistical analysis to relate the normalized 13 SNR (SRO) at deep HSL empirically to the MABL specific humidity (q) at 950 hPa (~400 m). However, the contribution of the paper is still marginal. Anyway, in order to improve the quality of the present manuscript, the following remarks should be addressed:
1. First of all, the organization of different sections are very poor as it is really hard to follow the sections.
2. The contribution of this paper is not clear i.e., what is the main contribution of the paper compared to the existing literature; the authors need to carefully describe the contributions of this work in the context of the literature. The main contribution of the paper shall be highlighted and emphasized. Moreover, please cite more recent papers to make a fair literature review.
3. Please explain the simulation and experimental platforms in more details so that the reader can easily understand your work.

4. Which AI/machine learning model is used in the work is not clear and needs to explained properly with some mathematical modelling.

5. The databases used in the study needs to be clearly mentioned with reference.

6. Which are the evaluation parameters used to access the performance of the proposed method?

7. Some references and text are not in the correct format. Moreover, there exist typos in the paper.

Reviewer 3 Report

This article presents a statistical analysis to relate the normalized signal-to-noise ratio at deep straight-line height empirically to the marine atmospheric boundary layer-specific humidity at 950 hPa. The content of the manuscript is well-organized and well-articulated. An extensive review of the literature has been conducted by the authors. The methodology presented by the authors is appropriate. The manuscript is well-written in general. The topic is of interest to the journal and related readers. The use of the English language is good. 

Line 27: ‘keyword 1’ should be omitted.

Lines 27-29: Please try to limit the keywords to 6.

Line 286: Please position the figure sub heading (a) under the figure and not on the left side of the figure.

Round 2

Reviewer 1 Report

well done